# Spatially Informed Autoencoders for Interpretable Visual Representation Learning

**Dominik Sturm**[1,2,3,4,*]**, Hiba Bensalem**[1,2,3,4,*]**& Ivo F. Sbalzarini**[1,2,3,4]

[1]Dresden University of Technology, Faculty of Computer Science, Dresden, Germany
[2]Max Planck Institute of Molecular Cell Biology and Genetics, Dresden, Germany
[3]Center for Systems Biology Dresden, Dresden, Germany
[4]Center for Scalable Data Analytics and Artificial Intelligence ScaDS.AI, Dresden/Leipzig, Germany
correspondence: sbalzarini@mpi-cbg.de

## Abstract

We introduce spatially informed variational autoencoders (SI-VAE) as self-supervised deep-learning models that use stochastic point processes to predict spatial organization patterns from images. Existing approaches to learning visual representations based on variational autoencoders (VAE) struggle to capture spatial correlations between objects or events, focusing instead on pixel intensities. We address this limitation by incorporating a point-process likelihood, derived from the Papangelou conditional intensity, as a self-supervision target. This results in a hybrid model that learns statistically interpretable representations of spatial localization patterns and enables zero-shot conditional simulation directly from images. Experiments with synthetic images show that SI-VAE improve the classification accuracy of attractive, repulsive, and uncorrelated point patterns from 48% (VAE) to over 80% in the worst case and 90% in the best case, while generalizing to unseen data. We apply SI-VAE to a real-world microscopy data set, demonstrating its use for studying the spatial organization of proteins in human cells and for using the representations in downstream statistical analysis.

## 1 Introduction

The spatial distribution of objects or events in images is an important readout in many applications. Examples include distributions of forest fires (Kato et al., 2020) or species abundance (Gillespie et al., 2024) in satellite imagery, or the distributions of viruses in biological cells observed by fluorescence microscopy (Helmuth et al., 2010). In these examples, semantic categories are not defined by appearance or texture of the imaged objects or events. The goal then is to infer representations of the observed spatial patterns that determine or explain them.

Deep learning has been particularly powerful to infer visual representations (Moen et al., 2019) and identify major sources of variation (Bengio et al., 2013). Microscopy images, for example, encode spatially structured patterns of discrete objects, such as cells in tissues or molecules in cells, that are causal for biological function and its dysregulation in disease (Hung & Link, 2011).

Several un- and self-supervised approaches have been proposed to extract information about spatial distributions in images. This includes contrastive learning by comparing augmented views of the same image (Chen et al., 2020). Contrastive learning, however, relies on pixel similarity and might therefore fail to capture spatial correlations. This was addressed by *Cytoself* in a domain-aware approach that used a classification pretext task to predict ground-truth protein labels (Kobayashi et al., 2022), constituting a semi-supervised approach (Kingma et al., 2014). Alternatively, protein sequences have been used to predict cellular localization (Khwaja et al., 2023; Kilgore et al., 2025). Such hybrid approaches model the joint density of the feature and label space, leading to useful representations (Nalisnick et al., 2019) leveraging ideas from predictive coding (Oord et al., 2019).

---

[*]Equal contribution.
  Code: https://git.mpi-cbg.de/mosaic/software/machine-learning/si-vae

In the absence of annotated ground truth, self-supervised autoencoders have shown promise for extracting biological features from microscopy images (Kraus et al., 2024). Usually, self-supervision is based on pixel intensities using masked autoencoders (He et al., 2022) or image transformations (Gatopoulos & Tomczak, 2021). Gaussian processes (GP) have been used to encode correlations in the data through structured priors (Casale et al., 2018; Pearce, 2020; Jazbec et al., 2021). This allows modeling correlations between images but not spatial correlations among objects within images. Recently, Vasan et al. (2025) proposed point clouds for representation learning of shapes and spatial distributions. The prediction likelihoods of point-cloud models structure the latent space to become discriminative about the spatial distribution. This is in contrast to class labels encouraging a categorical latent space and augmentations encouraging pixel-level similarity.

Such learned representations can be used to study biological associations and perturbations (Celik et al., 2022). They do, however, not provide mechanistic insight into spatial organization, and they lack a statistical framework for rigorous downstream analysis, which is a prerequisite for scientific applications. It has been shown that VAE can learn accurate surrogate models of GP priors for fast sampling (Semenova et al., 2022). While this accelerates spatial Bayesian inference, it does not model interactions between objects within an image. Recent work also revealed fundamental limitations of un- and self-supervised settings, such as the *Clever Hans* effect preferring bogus cues to true features (Kauffmann et al., 2025). Transformer-based masked autoencoders have been argued to primarily learn a representation based on the unmasked patches, ignoring the spatial arrangement of masked tokens in the decoder (Fu et al., 2025). This limits their ability to reason about spatial correlations between objects within an image. In addition to inherent limitations of the architectures, Gunawan et al. (2025) and Abgaryan et al. (2025) have recently shown that image metrics commonly used in loss functions focus on image appearance rather than spatial content. Together, these observations suggest that learning interpretable spatial representations from images requires additional priors.

Here, we propose spatial point processes as a self-supervision prior for visual representation learning. Spatial point processes are discrete stochastic processes from spatial statistics. As we show, they enable spatially informed variational autoencoders (SI-VAE) that learn statistically interpretable representations of spatial distributions of point-like objects in images. Spatial statistics has long been a powerful tool for analyzing localization patterns in images (Helmuth et al., 2010; Lagache et al., 2015; Summers et al., 2022). It provides interpretable and generative models, aligning with the growing interest in explainable models for biology (Chen et al., 2024; Rotem & Zaritsky, 2024). SI-VAE combine the mathematical rigor and statistical interpretability of spatial point processes with the approximation power of deep learning, where the latent representation acts as a predictor for the probability density of spatial distributions.

## 2 METHODS

SI-VAE learn spatially referenced representations from images by augmenting VAE with spatial point processes. We describe the VAE framework and introduce spatial point processes before combining them to the SI-VAE architecture in a common probabilistic framework. There, the VAE approximates the density of a spatial point process for modeling spatial distributions. This provides statistically interpretable models, as well as zero-shot generative models to sample from the estimated distribution.

### 2.1 VISUAL REPRESENTATION LEARNING USING VARIATIONAL AUTOENCODERS

We use VAE to learn representations from an unlabeled set of images $x = \{x_i\}_{i=1}^N, x_i \in \mathbb{R}^{W \times H \times C}$. A VAE is a generative model working under the assumption that the data $x$ can be reconstructed from a latent vector $z \in \mathbb{R}^l$ (Kingma & Welling, 2014). The aim is to maximize the likelihood of the data $x$ under the latent representation $z$, $p_\theta(x) = \int p_\theta(x|z)p(z)\,\mathrm{d}z$. Since this likelihood is intractable, VAE approximate the posterior distribution by a variational form, $q_\theta(z|x) \approx p(z|x)$. The variational form is commonly chosen as $q_\theta(z|x) = \mathcal{N}(\mu(\theta), \sigma(\theta)^2 \mathbb{I}_l)$. Here, $\mathcal{N}(\cdot)$ is a Gaussian with mean $\mu(\theta)$ and standard deviation $\sigma(\theta)$, predicted by an encoding neural network, samples of which can be used to approximate the evidence lower bound (ELBO) (Kingma & Welling, 2014):

$$\log p_\theta(x) \geq \mathbb{E}_q[\log p_\theta(x|z)] - \beta \mathrm{KL}(q_\theta(z|x) \,\|\, p(z)), \tag{1}$$

where $p(z)$ is a prior over the latent vector $z$, commonly chosen as $\mathcal{N}(0, \mathbb{I}_l)$, and $\beta$ weighs between the reconstruction and the prior (Higgins et al., 2017). The standard ELBO is obtained for $\beta = 1$. Maximimzing the ELBO approximately maximizes $p_\theta(x)$ and minimizes the Kullback–Leibler (KL) divergence between $q_\theta(z|x)$ and the true posterior $p(z|x)$ (Kingma & Welling, 2019). Therefore, samples from the variational distribution $z \sim q_\theta(z|x)$ (or quantities derived from them) can be used as representations for downstream analysis (Zhang et al., 2022). This, however, is limited to information contained in the posterior, possibly ignoring the second-order correlation structure between objects in the data.

## 2.2 SPATIAL POINT PROCESS MODELS

A spatial point process $X$ is a discrete stochastic process on $W \subseteq \mathbb{R}^d$, $d \geq 1$, where $X \subseteq W$ is an finite unordered set of points. Their distribution can be characterized by $N(B) = |X \cap B|$, i.e., the number of points in some subregion $B \subseteq W$ (Møller & Waagepetersen, 2003). Such distributions are difficult to characterize, except for the uncorrelated Poisson point process. It is therefore common to define the density of a point process relative to the unit-rate Poisson process (see Appendix A). This defines Gibbs point processes, which model second-order correlations as interactions between points, specified in terms of an *energy-based density*

$$p_\xi(X) \propto \exp\left\{ -\sum_{u \in X} \phi_\xi(u) - \sum_{\{u,v\} \subseteq X}^{\neq} \psi_\xi(u, v) \right\}. \tag{2}$$

Here, $\phi_\xi : \mathbb{R}^d \to \mathbb{R}$ and $\psi_\xi : \mathbb{R}^d \times \mathbb{R}^d \to \mathbb{R}$ are the first- and second-order potentials. In SI-VAE, they are represented by two separate shallow neural networks with parameters $\xi = (\xi_\phi, \xi_\psi) \in \Xi$. These potentials control the *a-priori* propensity of observing a point and the pair-wise interaction between points, respectively. Depending on $\psi_\xi(u, v)$, points can be attractive, neutral, or repulsive. This defines a general class of point processes, which we infer from data.

Maximum-likelihood estimation (MLE) of Gibbs processes from data is intractable due to the unknown normalizing partition function of the density $p_\xi(X)$. This is common to many energy-based models (Bengio et al., 2013; Gao et al., 2021; Tomczak, 2024). Therefore, we instead model the Papangelou conditional intensity (Ba & Coeurjolly, 2023)

$$\lambda_\xi(X, u) = \begin{cases} p_\xi(X \cup \{u\})/p_\xi(X) & u \notin X \\ p_\xi(X)/p_\xi(X \setminus \{u\}) & u \in X. \end{cases} \tag{3}$$

In an infinitesimal volume $\mathrm{d}u$ around $u$, $\lambda_\xi(X, u)\,\mathrm{d}u$ can be interpreted as the probability of observing a point at $u$ given all other points in $X$. For densities following equation 2, the Papangalou conditional intensity is $\lambda_\xi(X, u) = \exp\left\{-\phi_\xi(u) - \sum_{v \in X} \psi_\xi(u, v)\right\}$. This has been used to derive pseudo-likelihood estimators that provide unbiased estimating equations without the normalizing constant (Møller & Waagepetersen, 2007). The pseudo-likelihood approximates the MLE under a conditional independence assumption (Baddeley, 2007). While this can limit the statistical efficiency of the estimator, we still use pseudo-likelihood estimation here, as it provides a computationally efficient and statistically consistent objective for learning spatial point-process models from data. For a Gibbs process with conditional intensity $\lambda_\xi(X, u)$, the log-pseudo-likelihood is

$$\log \mathrm{PL}(\xi) = \sum_{u \in X \cap D} \log \lambda_\xi(X, u) - \int_D \lambda_\xi(X, u)\,\mathrm{d}u, \tag{4}$$

where $D = W \ominus R$ is an erosion of the domain $W$ by an interaction distance $R$ to avoid edge effects (Ba & Coeurjolly, 2023). We directly use equation 4 as a loss function for learning the Papangelou conditional intensity $\lambda_\xi(X, u)$ in a VAE.

## 2.3 SPATIALLY INFORMED VARIATIONAL AUTOENCODERS

We derive a VAE architecture that learns a latent image representation from which the Papangelou conditional intensity of a Gibbs point process, defined in equation 2, can be predicted that explains the observed point pattern in the image. While this self-supervision target is not limited to VAE architectures, a VAE enables uncertainty quantification of the model by resampling $q_\theta(z|x)$.

The pretext task for the VAE exploits point-process statistics with realizations $X$ obtained from images $x$ through, e.g., spot detection. Such weak labeling is commonly used for biomedical data (Yakimovich et al., 2021). The loss enforces the joint parameters of the model $q_\theta(z|x)$ to approximate the marginal $p_\theta(x)$ and the conditional probabilities $p_\xi(X|z)$ of a point process by maximizing the likelihood of the observed points $X$ with the latent vector $z$ as predictor:

$$\mathcal{L}(\theta, \xi) := \mathbb{E}_q[\log p_\theta(x|z)] + \mathbb{E}_q[\underbrace{\log p_\xi(X|z)}_{\approx \log \mathrm{PL}(\xi|z)}] - \beta \mathrm{KL}(q_\theta(z|x) \, \| \, p(z)). \quad (5)$$

This is illustrated in Fig. 1. The *inference model* $q_\theta(z|x)$ only relies on $x$, such that $X$ is only necessary during training, as usual in self-supervised VAE (Kobayashi et al., 2022). This estimation procedure generalizes the Bayesian variational formulation proposed by Zhou et al. (2022) for parametric models to non-parametrically estimating a model of the point process from images. Therefore, SI-VAE do not require (learning) an inference function from the complete point pattern to the model parameters, which is usually unavailable in practice. We approximate $\log p_\xi(X|z)$ by the log-pseudo-likelihood in equation 4 using $\lambda_\xi(X, u|z)$. The Gibbs potentials $\phi_\xi$ and $\psi_\xi$ are represented by two-layer neural networks with input $z$. We choose $\psi_\xi(u, v) = \psi_\xi(\|u - v\|_2)$ to be a symmetric, isotropic function. This ensures that the interactions between points are invariant under translation and rotation of the whole point pattern. As shown in Appendix B, models with more degrees of freedom for $\psi_\xi$ tend to converge to trivial solutions that do not account for the local interactions between points. We further constrain the model with distance-decaying weights $w_{uv}$ (equation 14) to learn local interactions of range $L$ with $\psi_\xi(u, v) = w_{uv}\psi_\xi(\|u - v\|_2)$. Since long-range interactions are indistinguishable from density inhomogeneity, this regularization is required for identifiability of inhomogeneous point processes. The interaction range $L$ is a hyperparameter (see Appendix C.2). Conceptually, this framework extends to anisotropic interactions by considering the (signed) difference between point positions, i.e., $\psi_\xi(u, v) = \psi_\xi(u - v)$ if sufficiently diverse directional data are available. The input to the interaction network then becomes a vector.

The loss in equation 5 admits a probabilistic interpretation as a hybrid model of $p(X, x) = p(X|x)p(x)$. Hybrid models $p(X, x)$ have been shown to learn richer and more outlier-robust representations than purely discriminative models (Nalisnick et al., 2019; Tomczak, 2024). We assume that $x \perp X|z$, such that $z$ captures all information, and that the variational posterior $q_\theta(z|x)$ provides a good approximation to $p(z|X, x)$. The first assumption is common in multimodal VAE (Wu & Goodman, 2018). It is not limiting in our setting, since the point pattern $X$ is deterministically obtained from the image $x$. Therefore, $z$ captures all relevant information about both modalities, trivially rendering them conditionally (on $z$) independent. The second assumption implies that the model should only rely on images for inference, which is fulfilled by design when learning visual representations. Under these assump-

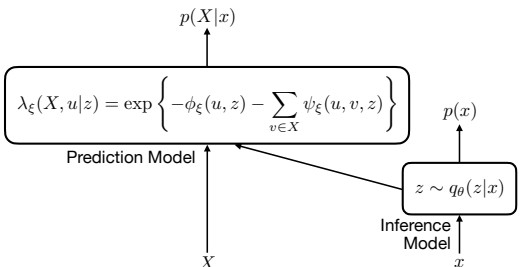

Figure 1: Schematic of the proposed SI-VAE architecture consisting of an *inference model* $q_\theta$ and a *prediction model* $\lambda_\xi$. The inference model uses an input image $x$ to sample the latent code $z$. The latent sample is used by the prediction model to predict the Papangelou conditional intensity $\lambda_\xi(X, u)$ of the point process $X$.

tions, the loss in equation 5 is the ELBO of a joint generative latent-variable model of the point process, see Appendix D for a proof[1]. This model has joint density $p(X, x, z) = p(X|x, z)p(x|z)p(z)$, where $p(x) = \int p(x, z) \, \mathrm{d}z$ is a standard VAE. The pseudo-likelihood in equation 4 can moreover be interpreted as the limit of Bernoulli random variables over partitions (or pixels) $u_i \subseteq W$, $i \in \{1, \ldots, I\}$ (Møller & Waagepetersen, 2007) with each partition/pixel conditioned on $X$:

$$\mathrm{PL}(\xi) = \lim_{\substack{|u_i| \to 0 \\ I \to \infty}} \prod_{1 \le i \le I} (\lambda_\xi(X, u_i|z)|u_i|)^{N_i} (1 - \lambda_\xi(X, u_i|z)|u_i|)^{1-N_i}. \quad (6)$$

The indicator $N_i = \mathbf{1}(N(u_i) > 0)$ denotes the presence of a point in partition $i$. Therefore, the *prediction model* $\lambda_\xi(X, u|z)$ can be interpreted as the limit of a pixel-wise classifier for the presence

---

[1]Note that the assumptions are only needed for the interpretation of the loss as an ELBO. They are not needed for SI-VAE to predict valid features of the Papangelou conditional intensity of a Gibbs process.

Table 1: Accuracy (Acc) and $F_1$ score for linear classification of point-pattern types in the latent space of the models (SI-VAE, VAE, mask VAE trained with perfect location knowledge) trained on images of different quality (SNR). We compare two different weak-labeling methods (*Spotiflow*, Thresholding) with a ground truth (GT) baseline using perfect knowledge of point locations. For the weak-labeling methods we also report the sensitivity $S_{F_1} = (F_{1,\text{GT}}^{\text{class}} - F_{1,\text{method}}^{\text{class}})/(1 - F_{1,\text{method}}^{\text{spot}})$ to quantify the robustness of the $F_1$ score to spot-detection errors.

| Model | SNR | GT Knowledge | | *Spotiflow* | | | Thresholding | | |
|---|---|---|---|---|---|---|---|---|---|
| | | Acc (↑) | $F_1$ (↑) | Acc (↑) | $F_1$ (↑) | $S_{F_1}$ (↓) | Acc (↑) | $F_1$ (↑) | $S_{F_1}$ (↓) |
| mask VAE | $\infty$ | 0.63 | 0.62 | × | × | × | × | × | × |
| VAE | 12.8 | 0.48 | 0.47 | × | × | × | × | × | × |
| VAE | 9.6 | 0.48 | 0.47 | × | × | × | × | × | × |
| SI-VAE | 12.8 | 0.90 | 0.90 | 0.90 | 0.90 | 0.0 | 0.80 | 0.80 | 0.56 |
| SI-VAE | 9.6 | 0.88 | 0.88 | 0.83 | 0.83 | 0.42 | 0.81 | 0.81 | 0.22 |

of a point given the data $X$. From Jensen's inequality, the second term in equation 5 provides a lower bound on the approximate conditional log-likelihood $\log p_{\xi,\theta}(X|x) \approx \log \mathbb{E}_q [p_\xi(X|z)] \geq \mathbb{E}_q [\log p_\xi(X|z)]$. This connects SI-VAE with other hybrid models, such as DIGLM (Nalisnick et al., 2019). SI-VAE, however, estimate $z$ using amortized variational inference instead of flow models. Therefore, SI-VAE learn a latent $z$ that approximates $p_{\xi,\theta}(X|x)$ while providing a model for the features $p_\theta(x)$. The details of the SI-VAE architecture used here, and of its training, are given in Appendix C.

## 3 EXPERIMENTS AND RESULTS

We benchmark SI-VAE on synthetic data, comparing them to VAE with the same architecture. The only difference between the VAE baseline and the SI-VAE is the presence of spatial supervision in the latter. This permits relative comparison. Then, we illustrate the workflow of applying SI-VAE to learning interpretable representations of protein localization patterns in human cells. We highlight introspection and interpretation of the latent space and show how the probabilistic framework of SI-VAE enables conditional simulation and downstream statistical analysis over learned representations.

### 3.1 LEARNING SPATIAL INTERACTIONS ON SYNTHETIC DATA

We first show that SI-VAE are able to learn representations that disentangle clustering due to attractive correlations from clustering due to inhomogeneous intensity functions. For this, we show that the latent space of an SI-VAE linearly separates homogeneous from inhomogeneous point processes and is able to classify their correlation structure.

**Data Generation**  We generate noisy synthetic images with known ground-truth point locations. Point locations are sampled from attractive (Thomas), repulsive (Strauss) and uncorrelated (Poisson) point processes with either homogeneous or inhomogeneous intensity functions. Appendix A details the benchmark point processes, Appendix E the synthetic image generation. It is important to note that Cox processes, such as the Thomas process, are not Gibbs processes, since their distribution is not defined via an energy function. Nevertheless, Gibbs processes constitute a flexible modeling framework, and our aim is to identify a Gibbs process that provides a suitable approximation to their distribution. We generate 5000 images for each of the six cases. All images have the same expected number of points $\mathbb{E}N(W) \approx 52$. This prevents the encoding network from simply discriminating between the processes by total pixel intensity.

**Experiment Design**  We compare the representation-learning performance of SI-VAE against a baseline VAE with the same architecture (see Appendix C.1) but without the point-process supervision target. Both models use the same encoder and decoder networks. The Papangelou conditional intensity $\lambda_\xi(X, u|z)$ is modeled using two separate two-layer neural networks for $\phi_\xi$ and $\psi_\xi$ (see

Appendix C.2) to predict the spatial distribution of $X$ from a given latent vector $z$. Both models are trained using the Adam optimizer (Kingma & Ba, 2015) over their respective ELBO until convergence on the validation set. We set $\beta = 0.1$, identified in a grid search over three orders of magnitude (0.001 to 1.0), and distance weight $w_{uv}$ with $L = 0.25$ determined from prior knowledge (see Appendix C.2). During evaluation, a representation is obtained as the mean of the posterior $q_\theta(z|x)$. We evaluate the representations obtained by both models using a linear evaluation protocol following Chen et al. (2020). This tests if $z$ contains enough information to distinguish the six different point processes in the data, thus linearly disentangling correlation from inhomogeneity. Since the performance of the proposed self-supervision task depends on the accuracy of the point patterns $X$ provided during training, we repeat the experiment for two different signal-to-noise ratios (SNR) using two different spot detectors (simple Otsu thresholding and the recent deep-learning method *Spotiflow* (Dominguez Mantes et al., 2025)). We compare them to a baseline with ground-truth (GT) knowledge of $X$.

**Results**   The results in Table 1 show that the SI-VAE consistently outperforms the standard VAE across all experiments. They also show that simple spot-detection methods, here Otsu thresholding, suffice to provide weak labels for the self-supervision task, especially at low SNR. When using state-of-the-art spot detection (*Spotiflow*, Dominguez Mantes et al. (2025)), the SI-VAE performs close to the GT-knowledge baseline. The SI-VAE predicts meaningful representations even at high noise levels (low SNR). We also report the degradation of the $F_1$ classification score w.r.t. the $F_1$ score of the spot detector, yielding the error sensitivity index $S_{F_1}$. In all cases, SI-VAE predictions degrade less quickly than spot-detection performance ($S_{F_1} \leq 1$), indicating that SI-VAE dampen spot-detection errors during training. As errors mainly occur by missing points in dense regions (see Table 3), however, they can bias the learned correlations (Kuronen et al., 2021).

Since SI-VAE incorporates structured knowledge in the form of (detected) point locations, we also compare it against a VAE trained on ground-truth binary object-location masks (mask VAE). This performs better than the VAE trained on whole images but does not reach the performance of SI-VAE. The UMAP projections of the latent spaces in Fig. 2 reveal that the standard VAE mainly learns global pixel-intensity patterns. This clearly manifests in the linear separation between homogeneous and inhomogeneous point processes, while not separating different process types (Strauss, Thomas, Poisson). While Semenova et al. (2022) have shown that VAE can learn spa-

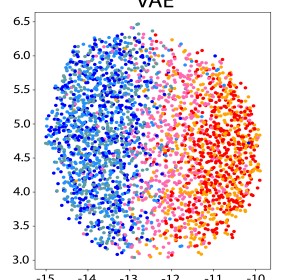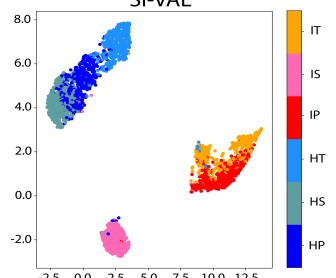

Figure 2: UMAP projections of the latent representations obtained by VAE (left) and SI-VAE (right) trained on images with an SNR of 12.8 and perfect GT knowledge of the point locations. Cold colors denote homogeneous processes (Poisson (HP), Strauss (HS), Thomas (HT)), warm colors inhomogeneous processes (Poisson (IP), Strauss (IS), Thomas (IT)).

tial correlations for GP priors, our results indicate that this is insufficient for explaining spatial point patterns. Only the SI-VAE distinguishes between process types, explaining the downstream classification accuracy. The latent space of the SI-VAE also maintains interpretability in the sense that mathematically more similar processes, like Poisson and Thomas, are mapped closer together than dissimilar processes (Strauss). This confirms that the proposed self-supervision task leads to more meaningful representations of the underlying spatial organization.

## 3.2 GENERALIZATION TO UNSEEN PROCESSES

In addition to testing the generalization of SI-VAE on a holdout data set, we quantify generalization to point processes of types not seen during training.

**Experiment Design**   We generate images from a homogeneous log-Gaussian Cox process (LGCP), which admits clustering through a Gaussian process (see Appendix A.4). Because of this clustering and definition based on a background process, it should behave similar to a homogeneous Thomas

process, which is a special case of a Cox process. We generate 150 synthetic images of point patterns from a LGCP with the same expected number of points as the other processes. We then use the linear classifier trained in the previous subsection on the (SI-)VAE representations from Table 1 to predict the class labels for the LGCP images. (SI-)VAE are provided with images only, without (extracted or ground truth) point locations.

**Results**   The results are shown in Fig. 3. The VAE baseline fails to generalize to the LGCP, as it mainly captures global pixel-intensity patterns. Even when trained on weak labels (colors, inset legends), the SI-VAE correctly classifies the LGCP to belong to the same class of processes as the Thomas process most similar to a homogeneous Thomas process (HT). The SI-VAE trained in Section 3.1 with complete knowledge of point locations classifies perfectly. Since the SI-VAE was not trained on LGCP data, this shows its capacity to generalize to unseen point processes, even when trained on weak labels.

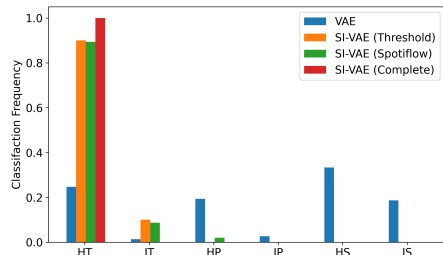

Figure 3: Class frequencies of the LGCP images predicted by the linear classifier on the representations trained in Table 1 for the higher SNR.

## 3.3   Interpreting the Learned Representations

SI-VAE provide representations that are interpretable in the framework of spatial statistics. The SI-VAE prediction $\lambda_\xi(X, u|z)$ approximates the Papangelou conditional intensity of a point process from an image. According to equation 3, the inferred point process is repulsive for $\lambda_\xi(X, u|z) \geq \lambda_\xi(Y, u|z)$, $X \subset Y$, and attractive otherwise. The predicted conditional intensities can also be visualized over the domain $D$. Finally, sampling multiple latent vectors $z$ from the posterior $q_\theta(z|x)$ enables uncertainty quantification of the estimates.

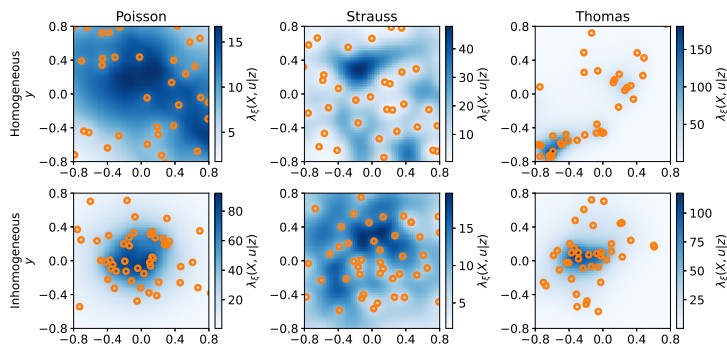

Figure 4: Visualization of the predicted conditional intensities $\lambda_\xi(X, u|z)$ for the latent codes $z$ obtained from different images $x$ from all considered point processes with homogeneous (top row) and inhomogeneous (bottom row) distributions over the eroded domain $D = W \ominus R$. Orange circles show the ground-truth point set $X$ for image $x$, which was not used during inference.

**Experiment Design**   We consider the SI-VAE from Section 3.1, trained with GT knowledge of the point locations at the higher image SNR. We randomly select images from the test set for each of the six point-process types. For each image, without knowledge of the point locations, the SI-VAE predicts the conditional intensity $\lambda_\xi(X, u|z)$.

**Results**   Figure 4 shows the predicted conditional intensities $\lambda_\xi(X, u|z)$ for different latent codes $z$ (additional plots in Fig. 11). Differences between homogeneous and inhomogeneous distributions are clearly visible. Moreover, the learned representations correctly capture local repulsion (Strauss) or clustering (Thomas) between the points, providing insight into the spatial organization directly from an image. Since SI-VAE model Gibbs processes, the learned potentials $\phi_\xi$ and $\psi_\xi$ can directly

Table 2: Relative intensity errors (RIE) of intensity estimates and rejection rates (RR) of second-order models for zero-shot conditional simulation of $\lambda_\xi(X, u|z)$ from the SI-VAE trained in Section 3.1. Average scores are computed over 100 random samples from the test set for each class.

| | Poisson | | Strauss | | Thomas | |
| --- | --- | --- | --- | --- | --- | --- |
| | RIE ($\downarrow$) | RR ($\downarrow$) | RIE ($\downarrow$) | RR ($\downarrow$) | RIE ($\downarrow$) | RR ($\downarrow$) |
| Homogeneous | 0.34 | 0.08 | 0.23 | 0.13 | 0.62 | 0.20 |
| Inhomogeneous | 0.27 | 0.03 | 0.14 | 0.14 | 0.49 | 0.15 |

be interpreted as the first- and second-order interaction structure of the process. For a Poisson point process, $\lambda_\xi(X, u) = \exp\{-\phi_\xi(u)\}$, such that $\psi_\xi(u, v) = 0$ for all $u, v$. This means that the SI-VAE should not predict any interactions between points. In Fig. 4, the SI-VAE predicts both attractive and repulsive interactions for the Poisson processes, depending on the latent code $z$. This suggests that the representations $z$ are sensitive toward the observed point configurations and unable to disentangle first- and second-order characteristics of the observed point pattern, which is a difficult task in general. Nevertheless, SI-VAE provide interpretable, mechanistic insight into learned representations. The interpretations, however, may be biased by specific point configurations observed in the image.

### 3.4 ZERO-SHOT CONDITIONAL SIMULATION FROM IMAGES

As shown in the previous experiment, SI-VAE learn Papangelou conditional intensities of point processes. Since $\lambda_\xi(X, u|z)$ parameterizes the distribution of $X$, it can be incorporated into an MCMC sampler (Møller & Waagepetersen, 2003), enabling conditional simulation. This provides a conditional (on a query image) generative model of the point process in addition to the generative model $p_\theta(x)$ for the images. Once trained, SI-VAE can thus explore the distribution of $X$ directly from an image $x$ in a *zero-shot* fashion. To our knowledge, this is the first instance of image-based conditional simulation for point processes.

**Experiment Design** We perform conditional simulation (see Appendix A.6) using the SI-VAE from Section 3.1, trained with GT point locations at high image SNR. For each point-process class, we select 100 random test images. We evaluate the simulations by comparing the predicted process $X_\xi$ to the observed $X$ under $\lambda_\xi$. Accuracy is assessed by nonparametric estimates of first- and second-order quantities: kernel density estimation (KDE) for intensity, and the $K$-function (Diggle, 2013) for interactions (see Appendix F). We compute the relative intensity error (RIE) between the KDE of observed and simulated point patterns, measuring over-/underprediction relative to the observed pattern. For interactions, we use a Monte Carlo test for point processes (Baddeley et al., 2014) and measure its rejection rate (RR), i.e., the frequency with which the $K$-function of the observed pattern significantly (5% significance level) deviates from the mean $K$-function.

**Results** Table 2 shows the RIE and RR obtained from the simulations. They confirm that the SI-VAE is able to generate point patterns close to the observed ones. The lowest RIE are obtained for the Strauss process, which is correctly predicted to be repulsive. In this case approximating $\lambda_\xi(X, u|z)$ works best and does not require model saturation (see Appendix A.6). As the envelopes in Fig. 12 show, the majority of hom. Poisson samples are also predicted to be repulsive. The higher RIE in this case is due to samples that appear clustering, producing excess points in the simulation. For estimates with non-clustering potentials we obtain good simulation results (Fig. 14). The RIE further increases for the Thomas process, which is always clustering. Here, model saturation (Appendix A.6) is crucial to obtain stable simulations. The error indicates that the model predicts too many points, which can also be seen in Figs. 14 and 15. This suggests that the saturation parameter $s$ is too large for these cases, motivating further investigation. The RR computed from the $K$-functions (Figs. 12 and 13) are overall low, indicating that the interaction structure is well captured. The lowest RR are obtained for the Poisson process, where the true interaction is always close to the conditioning sample when corrected by the normalization through the KDE (Fig. 13). The Strauss and Thomas processes have higher RR, which is expected due to the more complex interactions (repulsive, attractive). For these, Figs. 12 and 13 show that simulated point patterns exhibit weaker

interactions than the conditioning sample. Overall, however, the results show that the SI-VAE can be used for zero-shot conditional simulation of point processes, qualitatively capturing the correct correlation structure of the process from a single query image.

## 3.5 APPLICATION TO PROTEIN LOCALIZATION IN HUMAN CELLS

Having benchmarked SI-VAE on synthetic data, we next illustrate a workflow applying them to protein localization patterns in human cells. Specifically, we show that SI-VAE representations of cellular microscopy images distinguish between different localization patterns and that, thanks to their statistical interpretability, they allow for a meaningful biological interpretation.

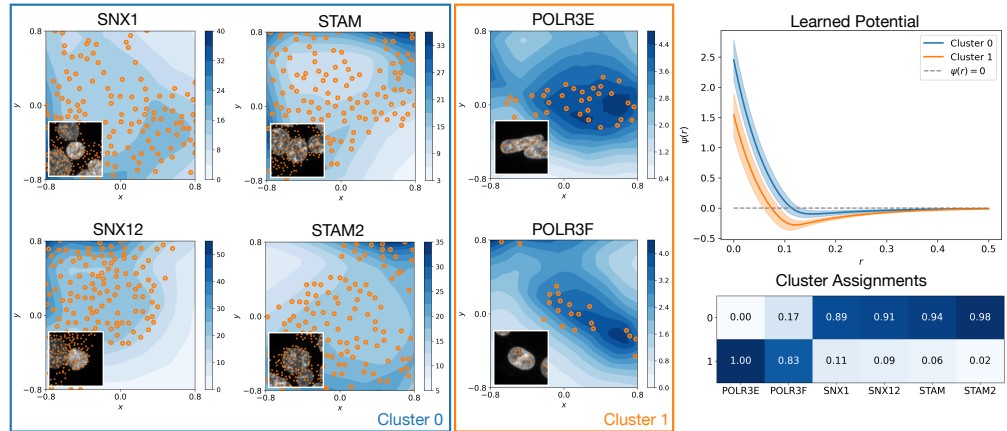

Figure 5: First-order (contour plots with detected point locations in orange) and second-order (top-right plot) Gibbs potentials of the SI-VAE representations on the test set. Insets show the nucleus channel for each sample as a reference. The two clusters (blue, orange) emerging in the latent space separate vesicle-localizing proteins from nuclear proteins with cluster-assignment frequencies for the six proteins given at the bottom right.

**Experiment Design** We train an SI-VAE on the *OpenCell* data set (Cho et al., 2022), comprising fluorescence microscopy images of the subcellular localization of over 1000 human proteins. We focus on six proteins from three families: four that localize to vesicles (SNX1, SNX12, STAM, STAM2) and two that form nuclear punctae (POLR3E, POLR3F). For SI-VAE training, we generate weak labels using *Spotiflow* (Dominguez Mantes et al., 2025). The SI-VAE is trained on 5499 images of the fluorescence signal of the tagged proteins, the nucleus channel, and a channel for the signed distance to the nucleus as previously described (Kobayashi et al., 2022). The full details are provided in Appendix G. We set the interaction length scale to $L = 0.25$, corresponding to the mean half-radius of the nuclei. We assess the learned representations on a disjoint test set containing 687 images of the same six proteins by clustering them with a Gaussian mixture model (GMM), selecting $K = 2$ clusters based on the Akaike information criterion (AIC) (see Appendix G). We compare the SI-VAE results with two baselines — a VAE with the same architecture but without spatial supervision and the specialized state-of-the-art *Cytoself* model (Kobayashi et al., 2022) with a substantially deeper architecture of two VQ-VAE (Oord et al., 2017) and a classification head.

**Results** Figure 5 shows the first-order $\rho_\xi(u) = \exp\{-\phi_\xi(u)\}$ and second-order $\psi_\xi(r)$ Gibbs potentials learned by the SI-VAE for the two GMM-identified clusters (cluster-assignment frequencies at the bottom right). The first cluster (blue) contains the four vesicle-associated proteins, while the second (orange) contains the two nucleus-localizing proteins. The GMM clusters obtained by the VAE and by *Cytoself* show the same result (Fig. 10). For the VAE, however, the clusters are diffuse with a Silhouette score of 0.04 (Table 4). SI-VAE and *Cytoself* obtain tighter clusters with Silhouette scores of 0.29 and 0.33, respectively, indicating that the learned representations more effectively capture differences in the images. This is confirmed by the UMAPs in Fig. 6, where SI-VAE obtains a good visual representation of the two protein families. The UMAP for VAE is similar to that in Fig. 2, suggesting that the VAE again mainly separates global intensity patterns. The tighter cluster-

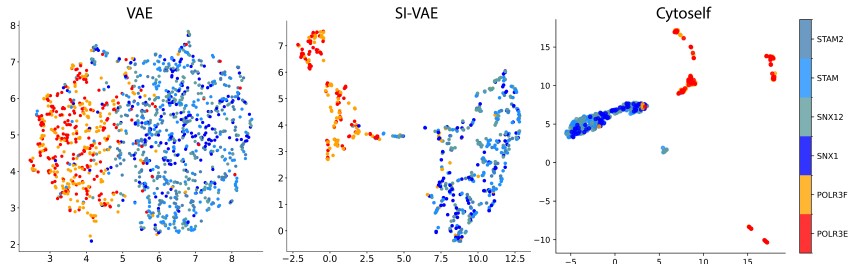

Figure 6: UMAP projections of the latent representations obtained by VAE (left), SI-VAE (middle), and *Cytoself* (right) trained on the *OpenCell* (Cho et al., 2022) human-protein data set. Cold colors denote vesicle-localizing proteins (SNX1, SNX12, STAM, STAM2), warm colors nuclear proteins (POLR3E, POLR3F).

ing of *Cytoself* is likely due to its deeper architecture and discrete latent space (vector quantization and classification labels as supervision target). SI-VAE achieves comparable clustering quality at much lower model complexity, while ensuring interpretability as spatial point processes. It also does not require ground-truth classification labels during training, enabling unbiased discovery.

The first-order potentials correctly reflect known localization patterns: proteins in vesicles are more homogeneously distributed, leading the SI-VAE to group them together, while nuclear proteins show inhomogeneous distributions localized to the nuclei. The two baseline models identify the same classes but do not explain them by an underlying probability distribution. The second-order SI-VAE potentials further distinguish the clusters according to the spatial interactions. They reveal short-range repulsion and long-range attraction. The repulsion has smaller range for nuclear than for vesicular proteins, explaining the tighter molecular packing in the nucleus. The nuclear proteins further show stronger long-range attraction, which facilitates efficient coverage of the nucleus, accelerating biochemical reactions in the diffusion-limited regime (Subic & Sbalzarini, 2024). This is corroborated by the conditional intensities $\lambda_\xi(X, u|z)$ predicted by the SI-VAE (Fig. 16) showing high event probabilities in the nuclei. We did not model or impose nuclear confinement. The SI-VAE reveals it from the data, providing a mechanistic explanation for the observed patterns.

## 4 CONCLUSION

We introduced a self-supervision target grounded in point-process statistics. The loss function of the resulting SI-VAE architecture can be understood as the ELBO of a joint model over images and point processes. This showed effective in capturing spatial interactions and generalizing to unseen data. The learned representations can directly be interpreted in the framework of spatial statistics. Since SI-VAE constitute hybrid models, we also demonstrated zero-shot conditional (on a query image) simulation of point processes. Unlike purely generative models (Lüdke et al., 2025; Zhou et al., 2022), SI-VAE remain interpretable. We highlighted the practical utility of SI-VAE by applying them to localization patterns of proteins in human cells. The SI-VAE correctly identified protein localization classes and provided a mechanistic explanation for their differences. This demonstrates that SI-VAE learn interpretable visual representations of spatial localization patterns.

In the future, SI-VAE could be extended to non-pairwise interactions by estimating the full density using, e.g., score matching (Hyvärinen, 2005; Cao et al., 2024) or Deep Sets (Zaheer et al., 2017). SI-VAE could also be extended to anisotropic interactions, if directionality is present in the data, and to marked point processes to model interactions across different types of points. Additionally, SI-VAE could be combined with more expressive priors on the latent variables, such as GP-VAE (Casale et al., 2018; Pearce, 2020; Jazbec et al., 2021) or VampPrior (Tomczak & Welling, 2018).

SI-VAE are straightforward to implement and train; they only require a small prediction model to be added to a VAE. The main additional cost during training is the evaluation of the Papangelou conditional intensity, which requires computing pairwise interactions between points and numerical quadrature. During inference, SI-VAE behave like regular VAE, as the point-process component is only used for training. This makes SI-VAE a drop-in replacement for VAE in applications where spatial localization patterns are an important feature.

ACKNOWLEDGMENTS

We thank Dr. Nandu Gopan and Dr. Abhishek Behera (both Sbalzarini group) for helpful discussions. We acknowledge financial support by the Federal Ministry of Research, Technology and Space of Germany and by the Saxon State Ministry of Science, Culture, and Tourism in the Center of Excellence for AI-research "Center for Scalable Data Analytics and Artificial Intelligence (ScaDS.AI)". This work was supported by the German Research Foundation (DFG) under Germany's Excellence Strategy, Cluster of Excellence "Physics of Life" (EXC2068).

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

# A  POINT PROCESS MODELS

We provide details of the point processes used in the synthetic benchmarks. All parameters were chosen such that all samples, independent of the process model and the homogeneity, have an expected number of $\mathbb{E}N(W) \approx 52$ points. We used the observation window $W = [-1, 1]^2$ for inference, whereas sampling was performed in extended domains $W_{\text{ext}}$ depending on the process, as discussed below, to avoid edge effects. For the inhomogeneous processes we used intensity trend functions of the form $\rho(u) = \kappa \exp\{-\|u\|_2^2/\alpha^2\}$. For each process we generated 5000 independent samples. More information on spatial point processes can be found in the popular monographs by Baddeley (2007), Møller & Waagepetersen (2007), and Møller & Waagepetersen (2003).

## A.1  POISSON PROCESS

The Poisson point process is the simplest model, in which spatial variations in the point density arise solely from intensity inhomogeneitites as there are no interactions between points. Since points in a Poisson process are pairwise uncorrelated, this provides a baseline that is fully characterized by the intensity function. The intensity function $\rho(u)$ of a point process is related to the first moment of the point distribution as $\mathbb{E}N(W) = \int_W \rho(u)\,\mathrm{d}u$. $X$ is a Poisson point process with intensity function $\rho(u)$ if and only if $N(B) \sim \mathcal{P}(\mathbb{E}N(B))$ for all $B \subseteq W$, and all $u \in X$ are *i.i.d.* in $W$ with

density $p(u) = \rho(u)/\mathbb{E}N(W)$. Here, $\mathcal{P}$ denotes the Poisson probability distribution. The density of a Poisson point process with respect to the unit-rate Poisson process ($\rho(u) = 1$) is

$$p(X) = \exp\left\{ -\int_W (\rho(u) - 1)\, du \right\} \prod_{u \in X} \rho(u). \tag{7}$$

Therefore, the Papangelou conditional intensity of a Poisson process is $\lambda(X, u) = \rho(u)$, such that the probability of observing a point at $u$ does not depend on $X$.

We sample realizations of a homogeneous Poisson process with $\rho(u) = 13$ and $N \sim \mathcal{P}(\mathbb{E}N(W))$ points uniformly over $W$. For the inhomogeneous Poisson case, we choose $\kappa = 67$ and $\alpha = 0.5$. The inhomogeneous samples are obtained by first sampling a homogeneous Poisson process with intensity $\rho_{\max} = \max_{u \in W} \rho(u)$, followed by independently thinning it using the thinning probability $\pi(u) = \rho(u)/\rho_{\max}$. This results in an inhomogeneous Poisson process with the desired intensity (Møller & Waagepetersen, 2003).

## A.2  THOMAS PROCESS

A Thomas process is a clustering process and a special case of a Cox process (Møller & Waagepetersen, 2003). It is described by a parent Poisson process $Y$ with intensity function $\rho(u)$. Conditional on each $y \in Y$, there is a child Poisson process $X_y$ with intensity

$$\rho_y(u) = \lambda \mathcal{N}(u; y, \sigma^2 \mathbb{I}_d), \tag{8}$$

where $\mathcal{N}(\cdot)$ is the $d$-dimensional normal distribution and $\lambda > 0$ controls the number of child points. The standard deviation $\sigma$ controls the spatial range (length scale) of the clustering. A Thomas point process is then defined by the superposition of all child point patterns $\cup_{y \in Y} X_y$. This also provides a straightforward way of sampling from a Thomas process by two nested realizations of a Poisson processes. Due to the dependence on the parent process $Y$, Thomas processes are said to be driven by a random field $Z(u) = \sum_{y \in Y} \rho_y(u)$. Following Coeurjolly et al. (2017), we write their density as

$$p(X) = \mathbb{E}\left[ \exp\left\{ -\int_W (Z(u) - 1)\, du \right\} \prod_{u \in X} Z(u) \right] = \mathbb{E}p(X|Z), \tag{9}$$

where $p(X|Z)$ is the density of $X$ conditioned on $Z$. The Papangelou conditional intensity follows as $\lambda(X, u) = \mathbb{E}p(X \cup \{u\}|Z)/\mathbb{E}p(X|Z)$, which is expected to increase in the vicinity of a point in $X$.

For our synthetic samples, we choose $\sigma = 0.1$ as the clustering range. We sample the Thomas process in a domain that is extended by $7\sigma$ in all directions in order to capture all interactions. For the homogeneous Thomas case, we use the parent intensity $\rho(u) = 4$ and $\lambda = 3$. In the inhomogeneous case, we use $\kappa = 23$, $\alpha = 0.5$, and $\lambda = 3$ and obtain the parent Poisson process through thinning as described in Appendix A.1. The number of child points for each cluster center $y \in Y$ follows $N_y \sim \mathcal{P}(\lambda)$ sampled *i.i.d.* in $W$ following $u \sim \mathcal{N}(y, \sigma^2 \mathbb{I})$.

## A.3  STRAUSS PROCESS

The Strauss process is a repulsive point process where points repel each other within a certain radius $R$. Unlike Poisson and Thomas processes, it is directly defined in terms of its Papangelou conditional intensity

$$\lambda(X, u) = \rho(u)\gamma^{\sum_{v \in X} \mathbf{1}(v \in B(u, R))}, \tag{10}$$

where $\rho(u) = \exp\{-\phi(u)\}$ corresponds to equation 2, $B(u, R)$ is the ball with radius $R$ centered at $u$ and $0 \leq \gamma \leq 1$ controls the strength of repulsion. For $\gamma = 0$ no points are allowed within a distance $R$ from any other point (hard-core process), whereas $\gamma = 1$ results in a Poisson point process where points do not interact at all.

Point processes directly defined by a conditional intensity are best sampled using MCMC (Møller & Waagepetersen, 2003). We here used the Metropolis–Hastings sampler from the `spatstat` package available in `R` (Baddeley et al., 2016).

For all samples, we chose $R = 0.2$ and $\gamma = 0.4$, which constitutes a strongly repulsive process. For the homogeneous case, we chose $\rho(u) = 40$, and for the inhomogeneous case $\kappa = 400$ and

$\alpha = 0.5$. We run each MCMC chain for $10^6$ iterations and use 200 uniformly random points to define the initial condition. The sample is obtained after the last iteration.

## A.4 LOG-GAUSSIAN COX PROCESS

A log-Gaussian Cox process (LGCP) is a clustering Cox process similar to a Thomas process. However, instead of using a parent Poisson process to define the driving random field $Z$, a LGCP uses a Gaussian process. Therefore, $Z = \exp\{Y\}$, where $Y \sim \mathcal{GP}(\mu, K)$ is a Gaussian process with mean function $\mu$ and covariance function $K$ (Møller & Waagepetersen, 2003). The density is the same as that of a Thomas process, albeit with this $Z$.

We sample a LGCP with $\mu = 1$ and Gaussian covariance function. The variance of the Gaussian covariance function is set to $\sigma^2 = 3.31$ and the scale parameter to 0.4. This LGCP is sampled using the `spatstat` package in R (Baddeley et al., 2016).

## A.5 SATURATION PROCESSES

The Strauss process defined in Appendix A.3 only possesses a valid density for $0 \le \gamma \le 1$. For $\gamma > 1$, the density is not integrable with respect to the unit-rate Poisson process. This implies that a Strauss process cannot be used to model clustering point patterns, as the parametric model might suggest. In fact, this property holds for all Gibbs processes, where purely attractive potentials lead to ill-defined densities. To avoid an unbounded number of events during simulation in such cases, we introduce a saturation parameter similar to a Geyer saturation process (Geyer, 1999; Rajala et al., 2018; Ba & Coeurjolly, 2023). Then, the density with respect to the unit-rate Poisson process is

$$p(X) \propto \exp\left\{ -\sum_{u \in X} \phi(u) - \frac{1}{2} \sum_{u \in X} \max\left[ -s, \sum_{\substack{v \in X \\ u \neq v}} \psi(u, v) \right] \right\}, \tag{11}$$

where $s \ge 0$ is a saturation parameter capping the interaction strength. The one-half prefactor originates from iterating over all non-unique point pairs. This recovers the density in equation 2 in the limit $s \to \infty$, since $\psi$ is symmetric. The saturation $s$ renders the density integrable, defining a valid point process that can also account for clustering. The Papangelou conditional intensity of this process is

$$\lambda(X, u) = \exp\left\{ -\phi(u) - \left( \Psi_s(u, X) + \sum_{v \in X} \Psi_s(v, X \cup \{u\}) - \Psi_s(v, X) \right) \right\}, \tag{12}$$

with Gibbs interaction term $\Psi_s(u, X) = \max\left[ -s, \sum_{\substack{v \in X}}^{u \neq v} \psi(u, v) \right]/2$.

In practice, $s$ needs to be estimated from the data. Since SI-VAE do not account for saturation, we estimate the saturation parameter *post hoc* by minimizing the MSE between the saturated and unsaturated models over $X \cap D$. This ensures that the true conditional intensity is approximately equal to the estimated one under saturation.

## A.6 ZERO-SHOT CONDITIONAL SIMULATION

To simulate from the learned conditional intensity $\lambda_\xi(X, u|z)$ we use a birth–death Metropolis–Hastings Algorithm (Alg. 7.4 in Møller & Waagepetersen (2003)). The sampler iteratively proposes to add or remove a point in the current configuration $X$ (with probability $p = 0.5$) using uniform proposal and death probabilities, i.e., $p_b(u) = 1/|D|$ and $p_d(u) = 1/|X|$, with acceptance determined based on $\lambda_\xi(X, u|z)$. We perform simulations in the eroded domain $D = W \ominus R$, with $R = 0.2$, to avoid bias outside the estimation domain. Each point is obtained after 30,000 iterations of the sampler. The obtained points are then eroded by $R$ again to avoid edge effects. We initialize the Markov chain with 52 uniformly distributed points in $D$.

In a Gibbs point process, the density can become ill-defined for attractive potentials, as it may become unbounded for increasing numbers of points. This can lead to instabilities during simulation. We avoid this by introducing a saturation parameter as previously suggested (Geyer, 1999; Rajala

et al., 2018) (see Appendix A.5). This caps $\lambda_\xi(X, u|z)$ beyond a certain number of observed points around $u$, ensuring that the density remains bounded and hence valid for simulation. The saturation parameter $s$ influences the model. We determine $s$ using grid search over $s \in [10^{-3}, 3.5]$ with 50 log-equidistant points such that the MSE between the saturated and unsaturated models over $X \cap D$ is always below $\epsilon = 10^{-3}$. This ensures that the saturated model is close to the original model, while avoiding the instability. We find this to work reliably in our benchmarks.

We simulate the Strauss process directly from the learned conditional intensity $\lambda_\xi(X, u|z)$, while for the Poisson and Thomas processes we use the saturated model to mitigate instabilities due to clustering.

## B  Approximating Conditional Intensities by Neural Networks

SI-VAE represent the Papangelou conditional intensity of a point process in Gibbs form using first- and second-order potential functions $\phi : \mathbb{R}^d \to \mathbb{R}$ and $\psi : \mathbb{R}^d \times \mathbb{R}^d \to \mathbb{R}$ (see equation 2). We restrict $\psi : \mathbb{R} \to \mathbb{R}$ to be a isotropic and symmetric in its arguments. This ensures that interactions between points are invariant under translation and rotation. While this choice is reasonable and widely used in spatial statistics (Møller & Waagepetersen, 2003), it does limit the space of functions an SI-VAE model can approximate.

In the following we will motivate this choice by a validation study considering different choices to model $\psi$. Specifically, we compare $\psi(u, v)$, $\psi(u-v)$, and $\psi(\|u-v\|)$. We use 150 samples from the homogeneous Strauss process as described in Appendix A.3. We train the three different models by minimizing the negative log-pseudo-likelihood from equation 4 over 500 iterations using batches of size 32 and the Adam optimizer (Kingma & Ba, 2015) with learning rate $5 \times 10^{-4}$. For these tests, we use deeper neural networks than those used in the main text (see Appendix C.2). Specifically, we here use 4 layers with 64 neurons each for both the first- and second order potential networks $\phi_\xi$ and $\psi_\xi$. This increased depth accounts for the missing information about the latent variable $z$ of the VAE inference model, which is absent in the present test. Overall then, the models have a similar number of parameters to the SI-VAE conditional intensities in the main text.

Figure 7 shows the predicted conditional intensities (CI) for a fixed point pattern $X$ for the different models. It can be seen that only the isotropic and symmetric model $\psi(\|u - v\|)$ is able to approximate the Papangelou conditional intensity of the observed point pattern (which, in this case, is a collection of overlapping disks). The other two models ($\psi(u, v)$, $\psi(u-v)$) fail to capture the locally repulsive behavior and instead predict an almost homogeneous conditional intensity. The magnitude of these conditional intensities corresponds to the intensity of a Poisson process (Appendix A.1).We think that this identifiability issue arises due to the conditional independence assumption of the pseudo-likelihood in equation 4, which does not sufficiently account for the local interactions between points. This causes the network to favor trivial solutions when given too much freedom in the form of the function to be learned. Constraining the second-order potential to be symmetric and isotropic sufficiently regularizes the second-order interaction structure in SI-VAE. This rationalizes the choice of an isotropic interaction function $\psi(\|u - v\|)$ in the main text.

## C  Architecture Details

We detail our (SI-)VAE architecture and the training procedure. The code, implemented in `pytorch` (Paszke et al., 2019), is available at `https://git.mpi-cbg.de/mosaic/software/machine-learning/si-vae`.

### C.1  VAE Architecture for the Inference Model

Both VAE and SI-VAE use the same convolutional variational autoencoder architecture to encode images into a 64-dimensional latent space and reconstruct them from there. This constitutes the *inference model* from Section 2.3. The encoder consists of three convolutional layers with kernel size 4, stride 2, and padding 1, progressively downsampling the images from $256 \times 256$ pixel to $32 \times 32$ while increasing the number of channels as $1 \to 16 \to 32 \to 64$. The output is flattened into a 512-dimensional vector and passed through a fully connected layer. Two linear layers then map this 512-dimensional vector to the mean $\mu$ and log-variance $\log \sigma^2$ of the 64-dimensional latent

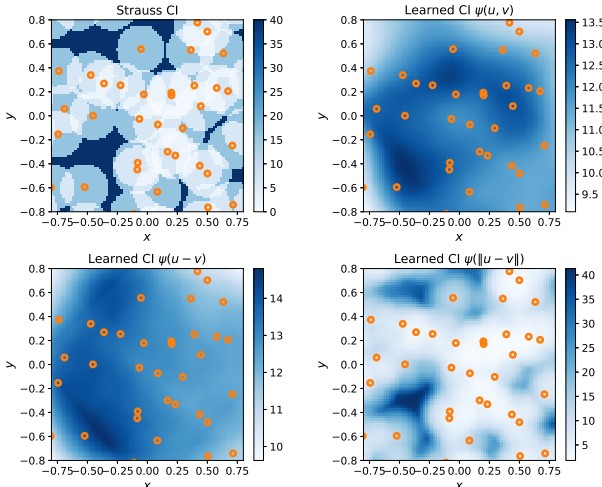

Figure 7: Comparison of different neural network architectures to model the conditional intensity (CI) $\lambda(X, u)$ of a homogeneous Strauss process. We compare the ground truth Papangelou conditional intensity of the Strauss process (top left) to predictions obtained from neural networks making no assumptions about the structure of the second-order potential ($\psi(u, v)$, top right), assuming it to be stationary ($\psi(u - v)$, bottom left), and assuming it to be symmetric and isotropic ($\psi(\|u - v\|)$, bottom right).

distribution ($512 \rightarrow 64$). Latent vectors $z$ are sampled using the reparameterization trick (Kingma & Welling, 2014). To reduce the variance, we average 100 samples per image to obtain $z$, effectively changing the latent distribution to $\mathcal{N}(\mu, \sigma^2/100)$ with the Kullback–Leibler divergence in the loss adjusted accordingly. We observed that this stabilizes training and improves convergence.

The decoder reconstructs images via a symmetric sequence of three transposed convolutional layers, progressively upsampling to the original image size, with channels decreasing as $64 \rightarrow 32 \rightarrow 16 \rightarrow 1$. All layers except the output layer use ReLU activations. The output layer generates the standardized image $\hat{x} \in \mathbb{R}^{W \times H \times C}$ without activations. As outlined in Appendix E.1, we treat pixel intensities as continuous and standardized. The autoencoders are thus trained by minimizing the mean squared error (MSE) between the input and reconstructed images in the reconstruction term of equation 1. Alternatively, pixel intensity levels could be treated as discrete categories, and the data-fitting term in the loss replaced with a categorical cross-entropy loss. All VAE models use $\beta = 0.1$, identified in a grid search over $[0.001, 1.0]$.

## C.2 ARCHITECTURE OF THE CONDITIONAL INTENSITY PREDICTION MODEL

In addition to the common inference model, SI-VAE additionally contain a *prediction model* as described in Section 2.3. The prediction model uses two neural networks, $\phi_\xi$ and $\psi_\xi$, to predict the conditional intensity function of a Gibbs process (cf. equation 2) as:

$$\lambda_\xi(X, u|z) = \exp\left\{-\phi_\xi(u, z) - \sum_{v \in X} w_{uv}\, \psi_\xi(\|u - v\|_2, z)\right\}, \tag{13}$$

where $z$ is the latent representation of the image $x$, $X$ is the corresponding point pattern, and $u \in W$ is the location at which the conditional intensity is to be predicted. The weights $w_{uv}$ emphasize local interactions between points to ensure model identifiability as discussed in Section 2.3. The network $\phi_\xi$ predicts the first-order potential $\phi_\xi(u, z)$ of the point process at location $u$ from the latent representation $z$. The network $\psi_\xi$ models the second-order potential $\psi_\xi(u, v, z)$ for the pairwise interaction between two points $u, v \in X$. Its inputs are a latent vector $z$ from the inference model and the Euclidean distances $\|u - v\|_2$. These pairwise interactions are aggregated in the sum over $v \in X$. Both $\phi_\xi$ and $\psi_\xi$ are implemented as separate two-layer neural networks, where the input is projected to 128 dimensions using a fully connected layer with ReLU activation, followed by another fully connected layer predicting the scalar potential value.

To ensure model identifiability as discussed in Section 2.3, the network $\psi_\xi$ emphasizes local interactions through exponential weights

$$w_{uv} = \exp\left\{-\frac{1}{2L^2}\|u - v\|_2\right\}, \tag{14}$$

with the hyperparameter $L$ informed by the interaction range in the data. For training, we compute the log-pseudo-likelihood for each pattern as in equation 4. The integral is numerically evaluated using the trapezoidal rule over a fixed grid of size $100 \times 100$. Edge correction is applied via domain erosion $D = W \ominus R$ with $R = 0.2$.

For each model variant, the three neural networks (VAE, $\phi_\xi$, $\psi_\xi$) are trained jointly, each with its own Adam optimizer (Kingma & Ba, 2015) using a fixed learning rate of $10^{-5}$ and a weight decay of $10^{-5}$. All SI-VAE variants were trained for 100 epochs, while the baseline VAE was trained for 150 epochs to achieve convergence. Early stopping was permitted to prevent overfitting.

## D    DERIVATION OF THE ELBO

For a paired set $\{(x_i, X_i)\}_{i=1}^N$ of unlabeled images $x_i \in \mathbb{R}^{W \times H \times C}$ and corresponding point patterns $X_i = \{u_j : u_j \in W \subseteq \mathbb{R}^d\}$, we aim to learn latent representations $\{z_i\}_{i=1}^N$ using a VAE with joint probability density $p(X, x, z) = p(X|x, z)p(x|z)p(z)$. We assume that $X \perp x|z$, i.e., that $X$ and $x$ are conditionally independent on $z$. We assume furthermore that $p(z|X, x)$ can be approximated by the inference model $q_\theta(z|x)$. While $X$ could hold additional information in practice, such as labels on the points, we require that during inference only $x$ is required. This is an architectural design choice. Under these assumptions, the evidence lower bound (ELBO) of the log-likelihood $\log p(X, x)$ can be derived as

$$
\begin{aligned}
\log p(X, x) &= \mathbb{E}_q\left[\log p(X, x)\right] \\
&= \mathbb{E}_q\left[\log\left(\frac{p(X, x, z)}{p(z|X, x)}\right)\right] \\
&= \mathbb{E}_q\left[\log\left(\frac{p(X, x, z)}{q_\theta(z|x)}\frac{q_\theta(z|x)}{p(z|X, x)}\right)\right] \\
&= \underbrace{\mathbb{E}_q\left[\log\left(\frac{p(X, x, z)}{q_\theta(z|x)}\right)\right]}_{=:\text{ELBO}} + \mathbb{E}_q\left[\log\left(\frac{q_\theta(z|x)}{p(z|X, x)}\right)\right].
\end{aligned}
$$

Since the KL divergence in the last term is always positive, we find a lower bound on the log-evidence $\log p(X, x)$. Moreover, maximizing the first expectation, which corresponds to the ELBO, minimizes the KL divergence between the variational distribution $q_\theta(z|x)$ and the true posterior $p(z|X, x)$. The quality of $q_\theta(z|x)$ thus determines how well the ELBO approximates the true log-likelihood. This implies that the inference model $q_\theta(z|x)$ has to be sufficiently expressive to approximate the true posterior well. This approximation could be improved by also using the point locations $X$ during inference. Here, however, we aim to learn visual representations, for which inference should only rely on images $x$. Focusing on the bound, the expression above can be further simplified to obtain equation 5 with $\beta = 1$:

$$
\begin{aligned}
\log p(X, x) &\geq \mathbb{E}_q\left[\log p(X, x, z) - \log q_\theta(z|x)\right] \\
&= \mathbb{E}_q\left[\log\left(p(X, x|z)p(z)\right) - \log q_\theta(z|x)\right] \\
&= \mathbb{E}_q\left[\log p(X, x|z)\right] + \mathbb{E}_q\left[\log p(z) - \log q_\theta(z|x)\right] \\
&= \mathbb{E}_q\left[\log p(X, x|z)\right] - \text{KL}(q_\theta(z|x)||p(z)) \\
&= \mathbb{E}_q\left[\log p(x|z)\right] + \mathbb{E}_q\left[\log p(X|z)\right] - \text{KL}(q_\theta(z|x)||p(z)).
\end{aligned}
$$

The last line relies on the conditional independence assumption.

The self-supervision target $\mathbb{E}_q[\log p(X|z)]$ provides a lower bound on the conditional log-likelihood $\log p(X|x)$. This follows from the joint model under the conditional independence assumption with

$p(X|x, z) = p(X|z)$. It then follows that

$$\log p(X|x) = \log \int p(X, z|x) \, \mathrm{d}z$$

$$= \log \int p(X|z)p(z|x) \, \mathrm{d}z$$

$$\geq \int \log p(X|z)p(z|x) \, \mathrm{d}z$$

$$\approx \int \log p(X|z)q_\theta(z|x) \, \mathrm{d}z = \mathbb{E}_q \left[ \log p(X|z) \right].$$

This shows that $q_\theta(z|x)$ is sufficient for predicting the point process $X$ conditioned on an image $x$.

The conditional independence assumption $X \perp x|z$ is an architectural design choice to simplify the model and the derivation of the ELBO. It postulates that $z$ captures the relevant information from both $X$ and $x$, such that once $z$ is known, knowing $X$ in addition to $x$ does not provide additional information about $x$, and vice versa. As a consequence, $p(X|x, z) = p(X|z)$, i.e., the point process $X$ can be predicted from the latent representation $z$ alone. This is desired, since we want $z$ to act as the predictor for the spatial distribution. Without this assumption, we would have to model $p(X|x, z)$ directly, which would require the image as a direct input to the prediction model, similar to conditional VAE (Sohn et al., 2015). The same conditional independence assumption is also used in multimodal VAE (Wu & Goodman, 2018). In practical applications of SI-VAE, the assumption should be fulfilled by construction, such that it is not limiting. Whenever the point locations $X$ are deterministically obtained from the image $x$, e.g., by spot detection (see Appendix E.3), or the images are synthetically generated from point locations (Appendix E), this is trivially the case. Both data modalities then capture the same underlying point process, and the conditional independence assumption is fulfilled.

# E    SYNTHETIC BENCHMARK DATA SET

We describe the procedure used to generate the synthetic benchmark data set used in the main text and how it was prepared for model training. The data sets are available at `https://git.mpi-cbg.de/mosaic/software/machine-learning/si-vae`.

## E.1    SYNTHETIC IMAGE GENERATION

We generate synthetic images that mimic the appearance of biological fluorescence-microscopy images. Images are generated from point process realizations by rendering each point as a disk-shaped region on a $300 \times 300$ pixel grid. Disk radii are sampled uniformly between 2 and 6 pixels to reflect the natural size variability of subcellular structures. All disks have a foreground intensity of 1.0 and the background intensity is 0.0 at the stage. The resulting binary disk image is convolved with a Gaussian kernel with standard deviation $\sigma_{\mathrm{blur}} = 4$ pixels, modeling the point-spread function of a fluorescence microscope (Zhang et al., 2007). A constant background intensity of 0.2 is added to simulate autofluorescence and ambient light. To realistically model photon shot noise, the blurred image with background is scaled by a gain factor of 100, converted to photon counts, and corrupted by Poisson noise with a rate proportional to the pixel intensity. This models detector shot noise. Subsequently, additive Gaussian noise with zero mean and standard deviation $\sigma_{\mathrm{read}} = 3$ is superimposed to model electronic amplification and readout noise. The final noisy image is normalized to $[0, 1]$ intensity range. An example of this process is shown in Fig. 8. The images were saved in PNG format with bit depth `uint8`. While this effectively discretizes the intensity levels, the discretization is fine enough to resolve the simulated photon-count levels. Therefore, we mathematically treat pixel intensities as continuous, thus images $x \in \mathbb{R}^{W \times H \times C}$.

To evaluate the impact of noise on representation learning, we also generated a data set with higher background intensity ($I_{\mathrm{bg}} = 0.4$). This changes the signal-to-noise ratio (SNR) of the images,

$$\mathrm{SNR} = \frac{I_{\mathrm{max}} - I_{\mathrm{bg}}}{\sqrt{I_{\mathrm{max}}}}, \tag{15}$$

where $I_{\mathrm{max}}$ is the maximum pixel intensity and $I_{\mathrm{bg}}$ is the background intensity. The numerator represents the maximum signal contrast, while the denominator measures the Poisson shot noise

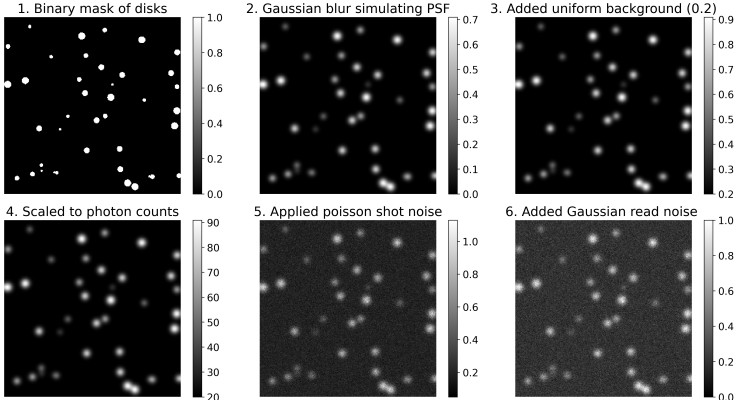

Figure 8: Example of the synthetic microscopy image generation pipeline for a spatial point pattern sampled from a homogeneous Poisson point process.

at the brightest pixel. The SNR of the images with the standard background ($I_{\text{bg}} = 0.2$) is 12.8, whereas it is 9.6 for the higher background ($I_{\text{bg}} = 0.4$).

## E.2 DATA SET PREPARATION

The data set consists of 5000 simulated images for each of the six point processes considered, paired with their respective ground-truth point clouds (see Section E.1). We split the data set into training (80%), validation (10%), and test (10%) sets. All images are scaled to $256 \times 256$ pixels to match the model input layer and normalized using the mean and standard deviation computed from the training set, with the same normalization applied to all splits to prevent data leakage. Batches of size 64 were used for training, with point coordinates zero-padded to the maximum number of points in the batch and an additional channel indicating the number of points.

## E.3 WEAK LABEL GENERATION

In a synthetic benchmark setting, the ground-truth point locations shown in an image are known. In practical applications, however, points first need to be detected in the images. This is only required for training an SI-VAE model. Inference can then be done on raw images.

We evaluate the robustness of SI-VAE to different point-detection methods and imperfect point detection. In addition to the ground-truth baseline, we therefore generate imperfect point locations as *weak* labels using two methods: (1) a simple global Otsu thresholding using `OpenCV` (Bradski, 2000) followed by contour centroid extraction, (2) the state-of-the-art deep-learning spot-detection method *Spotiflow* (Dominguez Mantes et al., 2025).

Before applying either spot-detection method, images were denoised using an isotropic Gaussian filter with standard deviation $\sigma = 3.0$ pixels. This value of $\sigma$ provided the best results. Smaller values left residual noise, whereas larger values caused too many misdetections.

Global Otsu thresholding was performed using `OpenCV` (Bradski, 2000) to obtain a binary mask. From this mask, we extracted the contour pixels of each connected component and retained all contours with nonzero enclosed area. For each retained contour we computed the geometric centroid (in $\mathbb{R}^2$). The centroid coordinates were then mapped to the normalized domain $[-1, 1]^2$ using standard Cartesian coordinates. This provides a simple and fast spot detector, which, however, tends to miss nearby spots and have low detection accuracy. It serves as a low-quality baseline for weak labeling.

To obtain high-quality weak labels of point positions, we used *Spotiflow* (Dominguez Mantes et al., 2025), a deep-learning spot-detection method specifically trained for fluorescence microscopy. It predicts the locations of bright spots in the image with subpixel resolution in $\mathbb{R}^2$. For each denoised image, we ran the pretrained *Spotiflow* model in subpixel mode, lowering the probability threshold to 0.40 to include candidates that might otherwise be rejected. This choice was informed by visual inspection of stereographic flow plots, where some spots were clearly visible but excluded under the

Table 3: Spot-detection performance for the two weak labeling methods (*Spotiflow* and Otsu thresholding) for images of different quality (SNR). We report the absolute difference (AD) between the ground truth and detected numbers of points, the True Positive Rate (TPR, recall), the Positive Predictive Value (PPV, precision), and the $F_1$ score. The expected number of points for all generated point patterns is $\mathbb{E}N(W) \approx 52$.

| Point process | SNR | *Spotiflow* | | | | Otsu thresholding | | | |
|---|---|---|---|---|---|---|---|---|---|
| | | AD ($\downarrow$) | TPR ($\uparrow$) | PPV ($\uparrow$) | $F_1$ ($\uparrow$) | AD ($\downarrow$) | TPR ($\uparrow$) | PPV ($\uparrow$) | $F_1$ ($\uparrow$) |
| Homogeneous Poisson | 9.6 | 6.68 | 0.874 | 0.999 | 0.932 | 19.03 | 0.637 | 1.000 | 0.777 |
| Homogeneous Poisson | 12.8 | 7.16 | 0.865 | 1.000 | 0.927 | 9.31 | 0.825 | 0.999 | 0.902 |
| Homogeneous Strauss | 9.6 | 4.32 | 0.918 | 0.999 | 0.956 | 15.60 | 0.701 | 1.000 | 0.824 |
| Homogeneous Strauss | 12.8 | 4.65 | 0.912 | 0.999 | 0.953 | 6.18 | 0.885 | 0.997 | 0.936 |
| Homogeneous Thomas | 9.6 | 14.99 | 0.732 | 0.999 | 0.843 | 29.38 | 0.466 | 1.000 | 0.633 |
| Homogeneous Thomas | 12.8 | 15.64 | 0.719 | 1.000 | 0.835 | 20.33 | 0.635 | 0.999976 | 0.772 |
| Inhomogeneous Poisson | 9.6 | 14.47 | 0.727 | 1.000 | 0.840 | 29.68 | 0.437 | 1.000 | 0.604 |
| Inhomogeneous Poisson | 12.8 | 15.28 | 0.712 | 1.000 | 0.830 | 20.06 | 0.622 | 0.999969 | 0.762 |
| Inhomogeneous Strauss | 9.6 | 6.7 | 0.870 | 0.999 | 0.929 | 18.81 | 0.634 | 1.000 | 0.775 |
| Inhomogeneous Strauss | 12.8 | 7.3 | 0.859 | 1.000 | 0.923 | 9.15 | 0.824 | 0.999168 | 0.901 |
| Inhomogeneous Thomas | 9.6 | 19.6 | 0.645 | 0.999 | 0.781 | 35.73 | 0.342 | 1.000 | 0.503 |
| Inhomogeneous Thomas | 12.8 | 20.57 | 0.627 | 0.999 | 0.767 | 27.10 | 0.508 | 0.999752 | 0.665 |

default adaptive probability threshold. The resulting coordinates were normalized to $[-1, 1]^2$ and mapped to standard Cartesian coordinates.

We quantify the quality of the two weak labeling methods—Otsu thresholding and *Spotiflow*—for the synthetic images for both image quality levels (SNR). For each image, we compare the detected points against the known ground truth. For this, we first use one-to-one Hungarian matching to compute a globally optimal correspondence between the two point clouds that minimizes the sum of squared distances between predicted and ground-truth points. A spatial cutoff of 3 pixels is applied to determine valid matches (Dominguez Mantes et al., 2025). This ensures that each predicted point is matched to at most one GT point, and vice versa. A matched pair is counted as a *true positive* (TP) if their Euclidean distance is below the cutoff. Unmatched predictions are counted as *false positives* (FP), and unmatched ground-truth points as *false negatives* (FN), while true negatives are undefined in this continuous-space setting.

For each point process type, we report the absolute difference (AD) between the ground truth number of points and the detected number of points, the True Positive Rate (TPR, recall), the Positive Predictive Value (PPV, precision), and the $F_1$ score across all 5000 images. The results are given in Table 3. As expected, *Spotiflow* consistently outperforms Otsu thresholding and is more robust to noise in the images (low SNR).

## F    SUMMARY STATISTICS AND SIMULATION METRICS

To validate the accuracy of the learned conditional intensity models, conditioned on a latent variable $z$, we compare nonparametric estimates of the first two moments of the predicted process and the observed point pattern. The first moment is the intensity function, defined as $\mathbb{E}N(W) = \int_W \rho(u)\, \mathrm{d}u$. It can be interpreted as the expected number of points per unit area. We obtain a nonparametric estimator using kernel density estimation (Møller & Waagepetersen, 2003). In particular, for an observed point pattern $X \subseteq W$, the intensity at location $u \in W$ is estimated as

$$\hat{\rho}(u) = \sum_{v \in X} \kappa_\eta(u - v)/w(v), \tag{16}$$

where $\kappa_\eta(u) = k(u/\eta)/\eta^2$ with density function $k$. We choose $k$ as the multivariate standard Gaussian and the weights $w(v) = \int_W \kappa_\eta(u - v)\, \mathrm{d}u$ to account for edge effects in the finite domain $W$ (Møller & Waagepetersen, 2003). We choose the bandwidth $\eta = 0.5$ empirically to yield good estimates for all considered point-process types. This is close to Scott's rule of thumb for $\mathbb{E}N(W) \approx 52$ points in $W = [-1, 1]^2$, which is $\eta_{\mathrm{scott}} = 0.518$ (Scott, 2015).

The accuracy of an intensity estimate is then quantified by the relative intensity error $\mathrm{RIE}(X_\xi, X) = n^{-1} \sum_{i=1}^n \int_W |\hat{\rho}(u|X_{i,\xi}) - \hat{\rho}(u|X)|/N(X)\,\mathrm{d}u$ for different samples $X_{i,\xi}$ obtained from the conditional intensity $\lambda_\xi(X, u|z)$ predicted by the SI-VAE for a given input image $x$.

To assess the quality of the estimated interaction structure, we leverage ideas from Monte Carlo goodness-of-fit tests for spatial point processes (Diggle, 2013; Baddeley et al., 2014). While such tests are strictly invalid and conservative for any significance level when parameters need to be estimated from the data, they provide a useful proxy to measure the quality of the learned model. We say that a model is not able to capture the interaction structure in the observed pattern if the observation $X$ lies outside the maximum envelope of the samples $X_{i,\xi}$ from the predicted conditional intensity $\lambda_\xi(X, u|z)$.

The sample envelope considers the minimum and maximum values of a functional summary statistic, such as the $K$-function, over multiple distances $r$. The $K$-function counts the normalized number of points at distance $r$. It is commonly used to characterize interactions in spatial point processes (Diggle, 2013). Following Baddeley et al. (2000), we estimate it as

$$\hat{K}(r) = \sum_{(u,v) \in X}^{\neq} \frac{\mathbf{1}(v \in B(u,r))}{\hat{\rho}(u)\hat{\rho}(v)e_{uv}}, \tag{17}$$

where $\hat{\rho}(u)$ is the estimated intensity at $u$, and $e_{uv}$ is an edge correction weight to account for the domain boundaries. We use translation-based edge correction as described by Møller & Waagepetersen (2003). The intensity function $\hat{\rho}(u)$ is estimated using equation 16. This shows that it can be difficult to disentangle errors in the first and second moments of the distribution, since both depend on $\hat{\rho}(u)$. This is especially true for clustering processes, where it is often unclear whether the clustering is due to a spatially varying intensity or due to attractive interaction between the points.

Since the functional summary statistic is a function of the $r$, we reduce it to a single test statistic $t_i$. Following Baddeley et al. (2014), we perform a maximum absolute deviation (MAD) test, which considers the maximum deviation from the mean $\overline{K}(r) = \frac{1}{n+1}\left(\hat{K}_1(r) + \cdots + \hat{K}_n(r) + \hat{K}_{\mathrm{obs}}(r)\right)$, where $\hat{K}_{\mathrm{obs}}(r)$ is the $K$-function of the conditioning sample and $\hat{K}_i(r)$ is the $K$-function of the $i$-th simulated sample. The test statistic then is

$$t_i = \max_{0 \le r \le R} |\hat{K}_i(r) - \overline{K}(r)|, \tag{18}$$

which fulfills the necessary symmetry property under the null hypothesis $H_0$ for a Monte Carlo test. We choose $R = 0.24$, which corresponds to the rule of thumb proposed by Diggle (2013). The rate of rejection of the test is used as a metric to assess the quality of the learned model. It quantifies how frequently the observed pattern $X$ achieves a larger maximum deviation from the mean than the samples $X_{i,\xi}$ from the learned model. Formally,

$$\mathrm{RR} = \frac{1}{M} \sum_{j=1}^M \mathbf{1}(\max_i t_{i,j} \le t_{\mathrm{obs},j}), \tag{19}$$

which measures the number of times the test rejects $H_0$. A high rejection rate (RR) indicates that the predicted conditional intensity significantly deviates from the observation and therefore insufficiently captures the interaction structure in the image. This is in line with the *post-hoc* model evaluation typically done for spatial point processes (Møller & Waagepetersen, 2007). We use $M = 100$ random images for each class from the test set and $n = 19$. This would correspond to a test significance level of $5\%$ for a one-sided test, if the parameters of the null model were known.

## G  APPLICATION TO SUBCELLULAR PROTEIN LOCALIZATION

We describe the data set preparation, model architecture, and training procedure for the protein localization experiment of Section 3.5.

We used the publicly available *OpenCell* data set (Cho et al., 2022), which contains fluorescence microscopy images of over 1000 human proteins. The present SI-VAE model was trained on images of size $100 \times 100$ pixels, each centered around a nucleus. Three channels were used for training: the

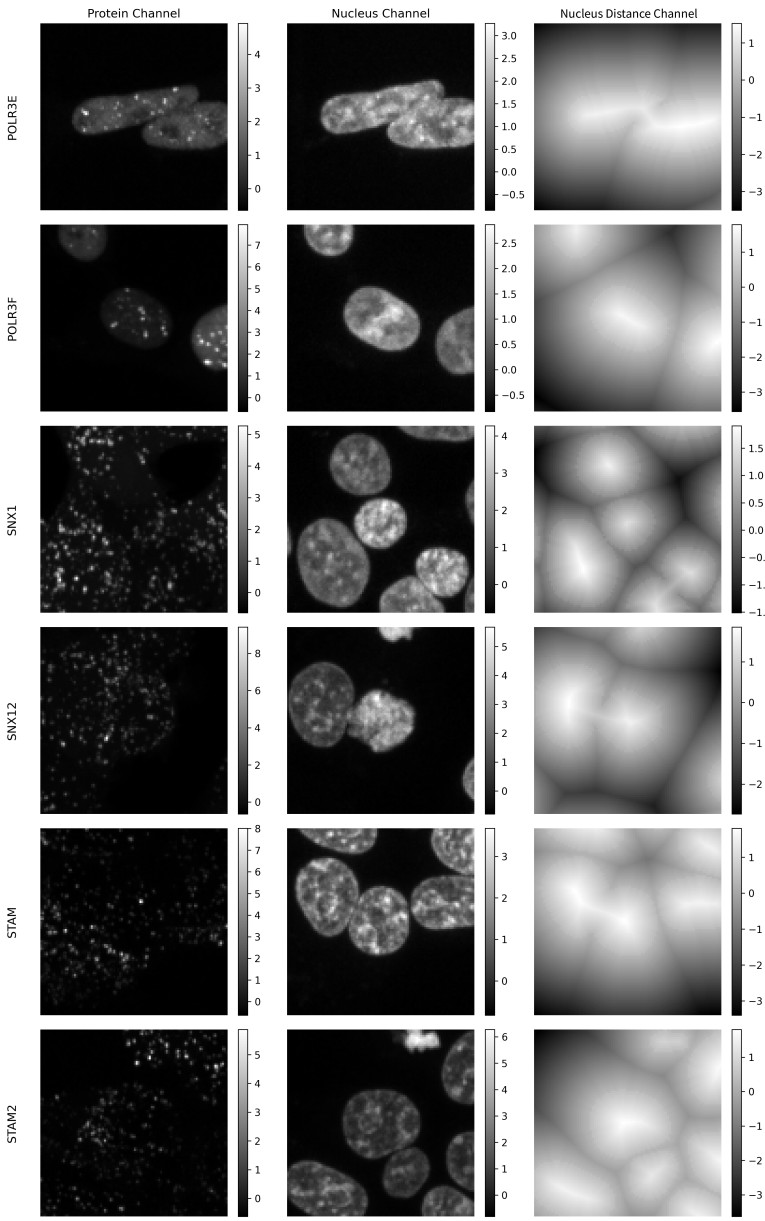

Figure 9: Representative standardized images from the test set for the six proteins (rows) considered in Section 3.5. Each image consists of three channels: the protein fluorescence signal (first column), the nucleus fluorescence signal (second column), and the signed distance function to the nucleus (third column). The same example images are also used in Figs. 5 and 16.

protein fluorescence signal, the nucleus fluorescence signal, and the signed distance function to the nucleus (see Fig. 9). Centering the observation window around a nucleus ensures that at least one cell is completely contained within an image. Since the same data preparation was used for *Cytoself* (Kobayashi et al., 2022), we directly took the images from the *Cytoself* repository (Royer Lab, 2025). We generated weak labels for protein spot locations using *Spotiflow* (Dominguez Mantes et al., 2025) on the protein fluorescence channel, with a low probability threshold of 0.35 to include candidate points that might otherwise be rejected. No Gaussian smoothing was applied to the images prior to spot detection. This yielded better results upon visual inspection. The resulting point

coordinates were normalized to $[-1, 1]^2$ and mapped to standard Cartesian coordinates. These point coordinates were then used as self-supervision target when training the SI-VAE model.

For the application case presented in the main text, we chose six proteins from three families. For each protein, about 1000 images were available (POLR3E: 1162, POLR3F: 1068, SNX1: 1001, SNX12: 1260, STAM: 1361, STAM2: 1021). This resulted in a data set of 6873 images, which was split into disjoint training (80%=5499 images), validation (10%=687 images), and test (10%=687 images) sets. Each channel in the training data set was standardized to zero mean and unit variance. The same standardization was then also applied to the validation and test sets to prevent data leakage. Representative images for each of the six proteins are shown in Fig. 9. These are the same representative examples as in Figs. 5 and 16.

The VAE architecture from Appendix C.1 was adjusted to the image shape of this application (100×100×3). The input channel size was set to 3, while the rest of the convolutional layers retained the same parameters as for the synthetic benchmarks, progressively downsampling the images to 12×12 while increasing the number of channels as 3→16→32→64. All convolutional layers used ReLU activations. The latent space dimension remained 64. The decoder reconstructed images via a symmetric sequence of three transposed convolutional layers, decreasing the number of channels as 64→32→16→3. All transposed convolutional layers used the same parameters as in Appendix C.1, with ReLU activations in all layers except the output, to predict the standardized images. The reconstruction error in equation 1 was minimized using the mean squared error, with the KL divergence scaled by $\beta = 0.1$ as in the synthetic benchmarks.

The conditional intensity prediction model remained unchanged from the synthetic benchmarks, as described in Appendix C.2. We set the range of the interactions to half the average nucleus radius, $L = 0.25$. This radius was identified from the central nuclei in all cropped images by taking the largest connected component in the nucleus channel and computing its equivalent diameter (diameter of a circle with the same area). This resulted in an average (across the entire data set) half-nucleus radius of 12.59 pixels. Transformed to the coordinate space $[-1, 1]^2$ used by the model, this yields $L = 0.25$. As in the synthetic benchmark, we used an erosion of $R = 0.2$ to avoid edge effects and discretized integrals over the domain using trapezoidal quadrature on a fixed grid of size 100×100.

Like in Appendix C, all three neural networks (VAE, $\phi_\xi$, $\psi_\xi$) were trained jointly on the training set, each with its own Adam optimizer (Kingma & Ba, 2015), using batches of size 16 and learning rate $10^{-4}$. Training stopped upon convergence of the validation loss.

We compared SI-VAE with two baselines: the VAE without spatial supervision and *Cytoself* (Kobayashi et al., 2022), a semi-supervised model based on vector-quantized VAE (VQ-VAE) that leverages ground-truth protein labels as self-supervision target. *Cytoself* constitutes the state of the art in human protein localization modeling, and it is significantly more complex than the present SI-VAE.

For the VAE baseline, we retained the SI-VAE's VAE component alone with the same parameters (64 latent dimensions, $\beta = 0.1$). Training was stopped upon convergence of the validation loss to ensure comparability.

For *Cytoself*, we used the published implementation (Royer Lab, 2025) but retrained the model from scratch on our data set. To correct for the smaller size of our data set, we reduced both VQ-VAE codebooks to 128 entries (from the original 512). The two VQ-VAE learn representations on different scales: a local representation (VQ1) at 25×25 pixel capturing texture detail and a global representation (VQ2) as a 4×4 pixel feature map with 64 features per pixel aimed at capturing long-range patterns. Following the original *Cytoself* paper, downstream clustering was performed on the global representation (VQ2). Training was performed with an initial learning rate of $10^{-4}$, which was reduced ten-fold whenever the validation loss did not improve for four consecutive epochs. Early stopping was applied with 10 epochs patience. Training terminated after 14 epochs.

All learned representations were analyzed using images from the test set. We determined the latent representation $z$ of each test image using the trained SI-VAE, VAE, and *Cytoself* encoders. For SI-VAE and VAE, we used the mean prediction of the posterior $q_\theta(z|x)$. We then standardized the latent representations across the test set to zero mean and unit variance for each latent dimension and performed clustering using a Gaussian Mixture Model (GMM) with full covariance matrix (Hastie et al., 2009). The number of clusters $K$ was determined using the Akaike Information Criterion

(AIC) (Akaike, 1974), $\text{AIC} = -2\log\hat{\ell} + 2K$, where $\hat{\ell}$ is the maximized likelihood of the model. We fitted GMM for $K = 1, \ldots, 6$ and selected the model with the lowest AIC, which was $K = 2$. We used the `scikit-learn` implementation of GMM (Pedregosa et al., 2011) with default settings. Silhouette scores (Rousseeuw, 1987) were computed using Euclidean distance on the serialized (into a vector) 64-dimensional embeddings (1024-dimensional for *Cytoself*). This score quantifies the average distance between points within the same cluster relative to the points in the nearest neighboring cluster. Higher Silhouette scores indicate tighter and more separated clusters. The Silhouette score for a single sample is defined as

$$S_c = \frac{b - a}{\max(a, b)},\tag{20}$$

where $a$ is the mean intra-cluster distance, and $b$ is the mean nearest-cluster distance. Scores range from $-1$ to $1$.

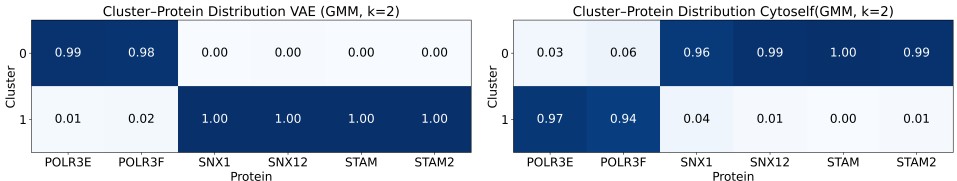

Figure 10: Cluster assignment frequencies for all proteins in the VAE (left) and *Cytoself* (right) embeddings. Each square shows the fraction of images of a given protein that was assigned to the corresponding cluster. Clusters were determined by Gaussian Mixture Models (GMM) with $K = 2$.

Table 4: Silhouette scores on the test set embeddings of each model. Higher values indicate more compact and better-separated clusters. The table reports the average Silhouette score for each cluster (Nuclear, Vesicular) as well as the overall score across all samples.

| Model | Overall | Vesicular cluster | Nuclear cluster |
|---|---|---|---|
| *Cytoself* | 0.331 | 0.227 | 0.550 |
| SI-VAE | 0.294 | 0.286 | 0.310 |
| VAE | 0.040 | 0.042 | 0.035 |

All three models consistently identified the same clusters (see Fig. 5 for SI-VAE and Fig. 10 for VAE and *Cytoself*). SI-VAE and *Cytoself*, however, achieved significantly higher Silhouette scores than the VAE baseline (Table 4). Since the cluster indices assigned by the GMM differ across models, each cluster in the table is labeled according to its dominant protein localization: nuclear (POLR3E, POLR3F) or vesicular (SNX1, SNX12, STAM, STAM2).

# H ADDITIONAL FIGURES

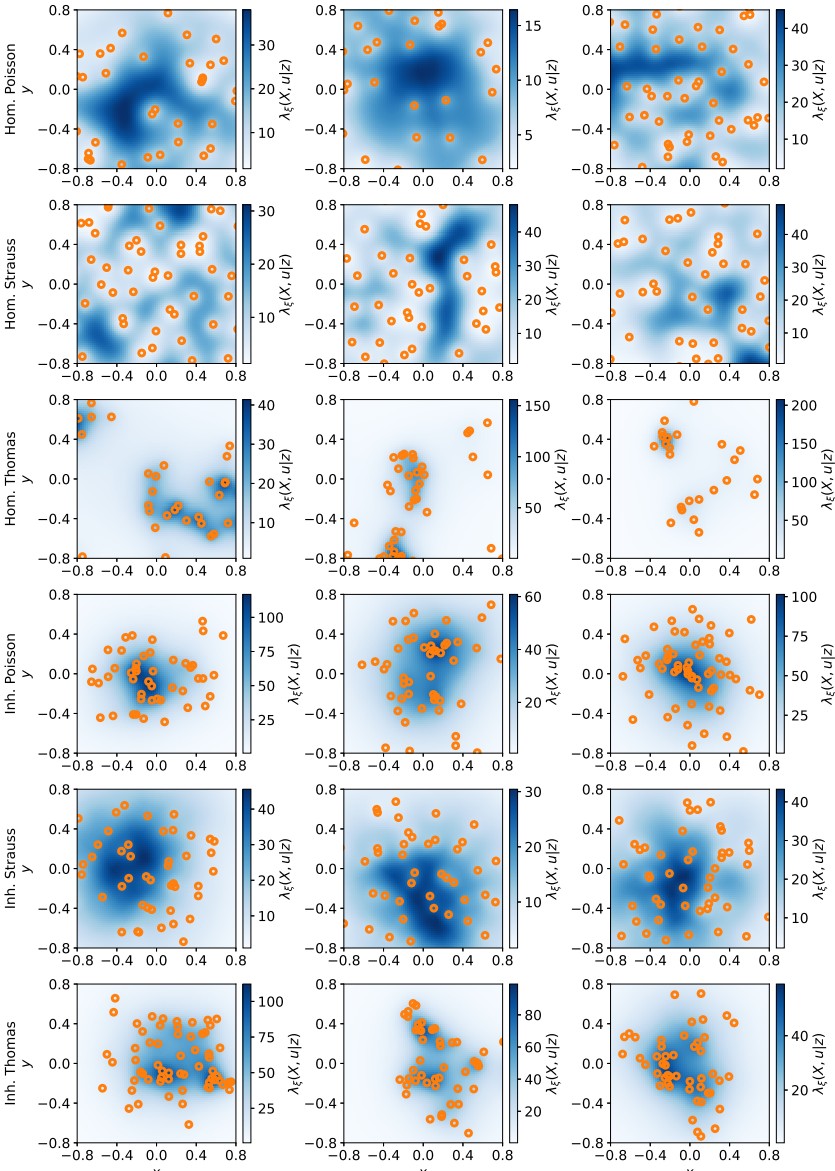

Figure 11: Visualization of predicted conditional intensities $\lambda_\xi(X, u|z)$ for different latent representations $z$ predicted by the SI-VAE from Section 3.1, trained at high SNR with ground-truth knowledge. Each row shows the conditional intensity for samples from all considered point processes (row labels on the left) over the eroded domain $D = W \ominus R$. For each point process, three *i.i.d.* examples are shown (columns).

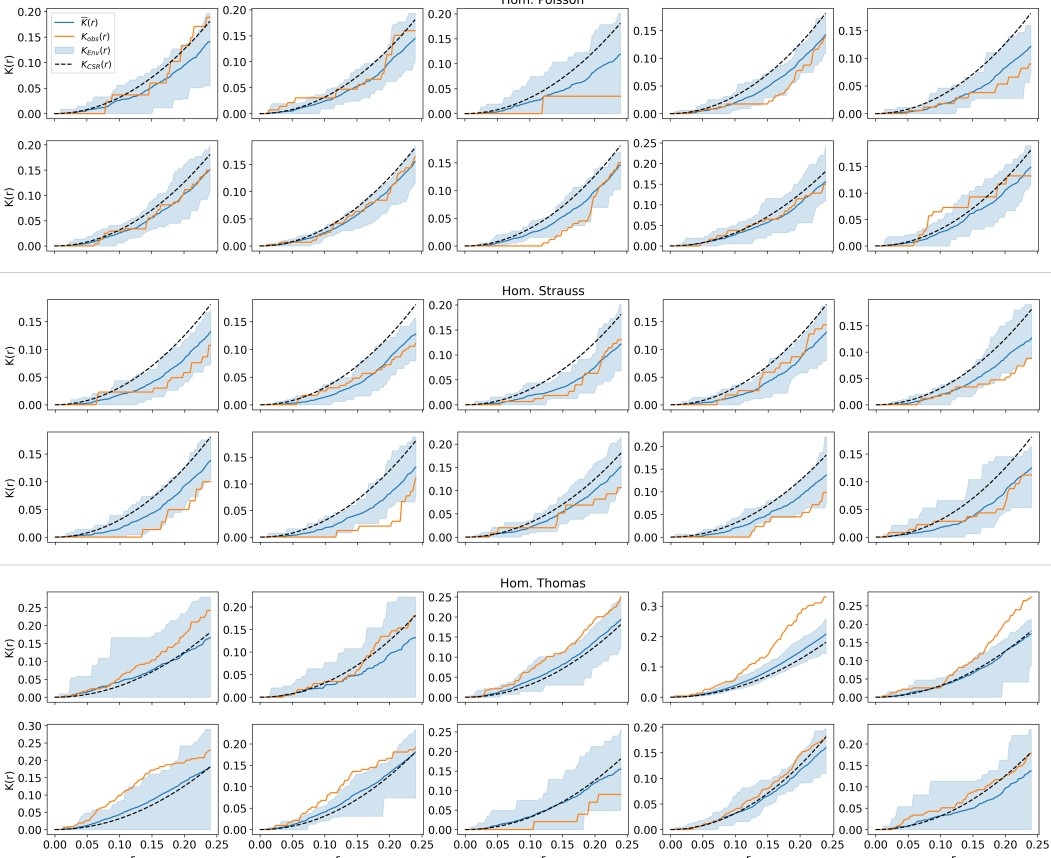

Figure 12: Simulation envelopes of the $K$-function for different latent codes $z$ in the homogeneous cases over the domain $[-0.6, 0.6]^2$. The orange line shows the empirically observed $K$-function, the blue area the maximum and minimum point-wise simulation envelopes, and the black dashed line the theoretical $K$-function for a Poisson process. The simulation envelopes are obtained from the learned representation of the SI-VAE from Section 3.1, trained at high SNR with ground-truth knowledge.

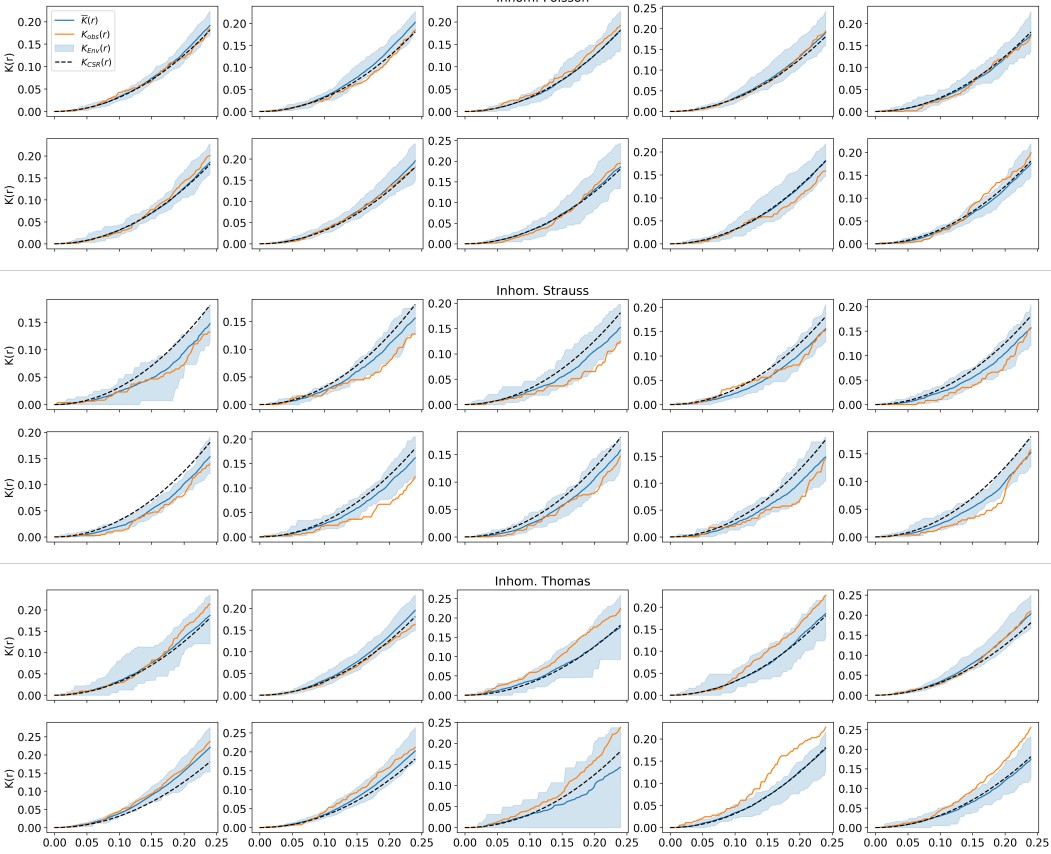

Figure 13: Simulation envelopes of the $K$-function for different latent codes $z$ in the inhomogeneous cases over the domain $[-0.6, 0.6]^2$. The orange line shows the empirically observed $K$-function, the blue area the maximum and minimum point-wise simulation envelopes, and the black dashed line the theoretical $K$-function for a Poisson process. The simulation envelopes are obtained from the learned representation of the SI-VAE from Section 3.1, trained at high SNR with ground-truth knowledge.

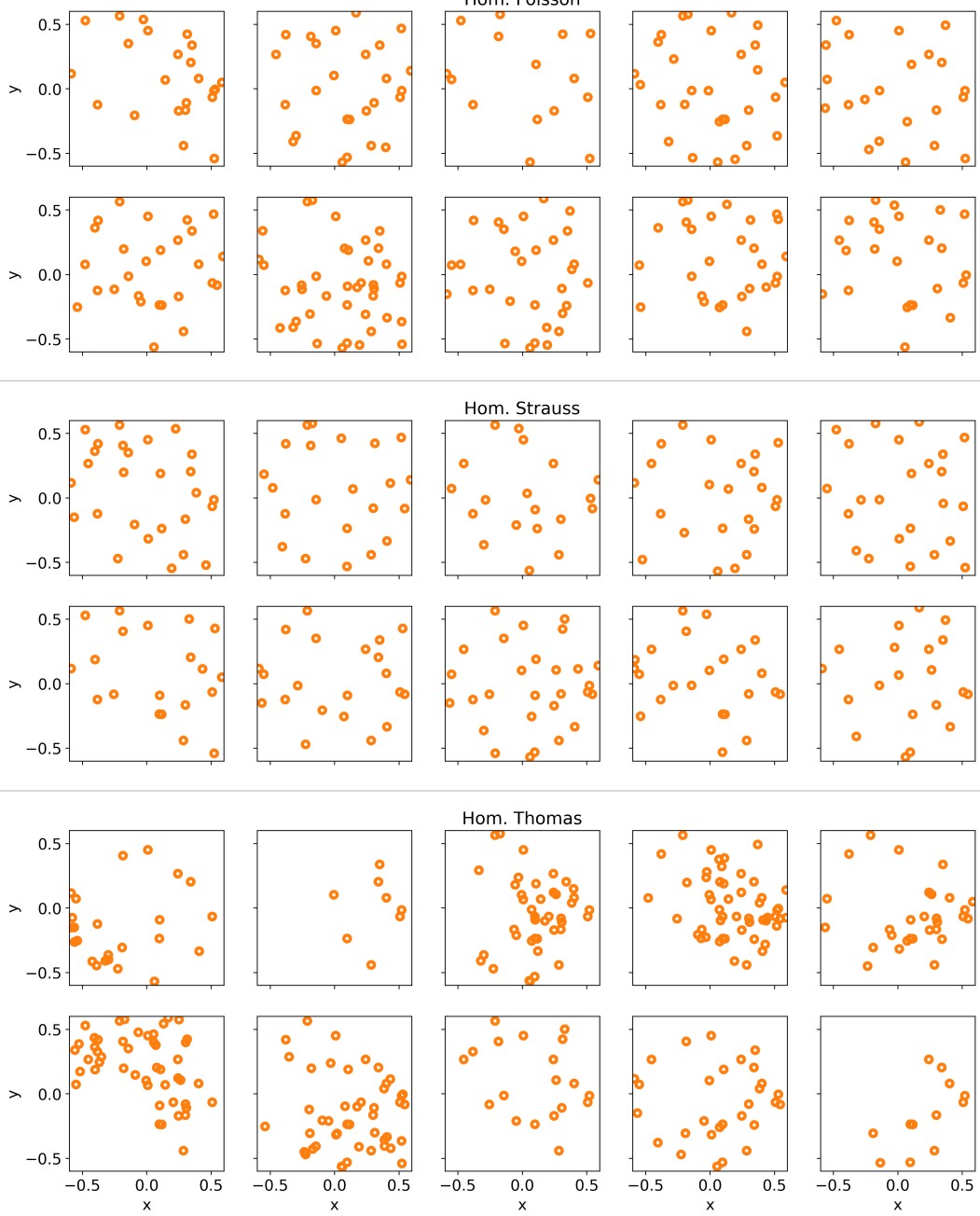

Figure 14: Examples of conditional simulation results from the trained conditional intensity model for different latent codes $z$ in the homogeneous cases over the evaluation domain $[-0.6, 0.6]^2$. The trained conditional intensity model is from the SI-VAE from Section 3.1, trained at high SNR with ground-truth knowledge. Ten *i.i.d.* samples are shown for each point process type (Poisson, Strauss, Thomas).

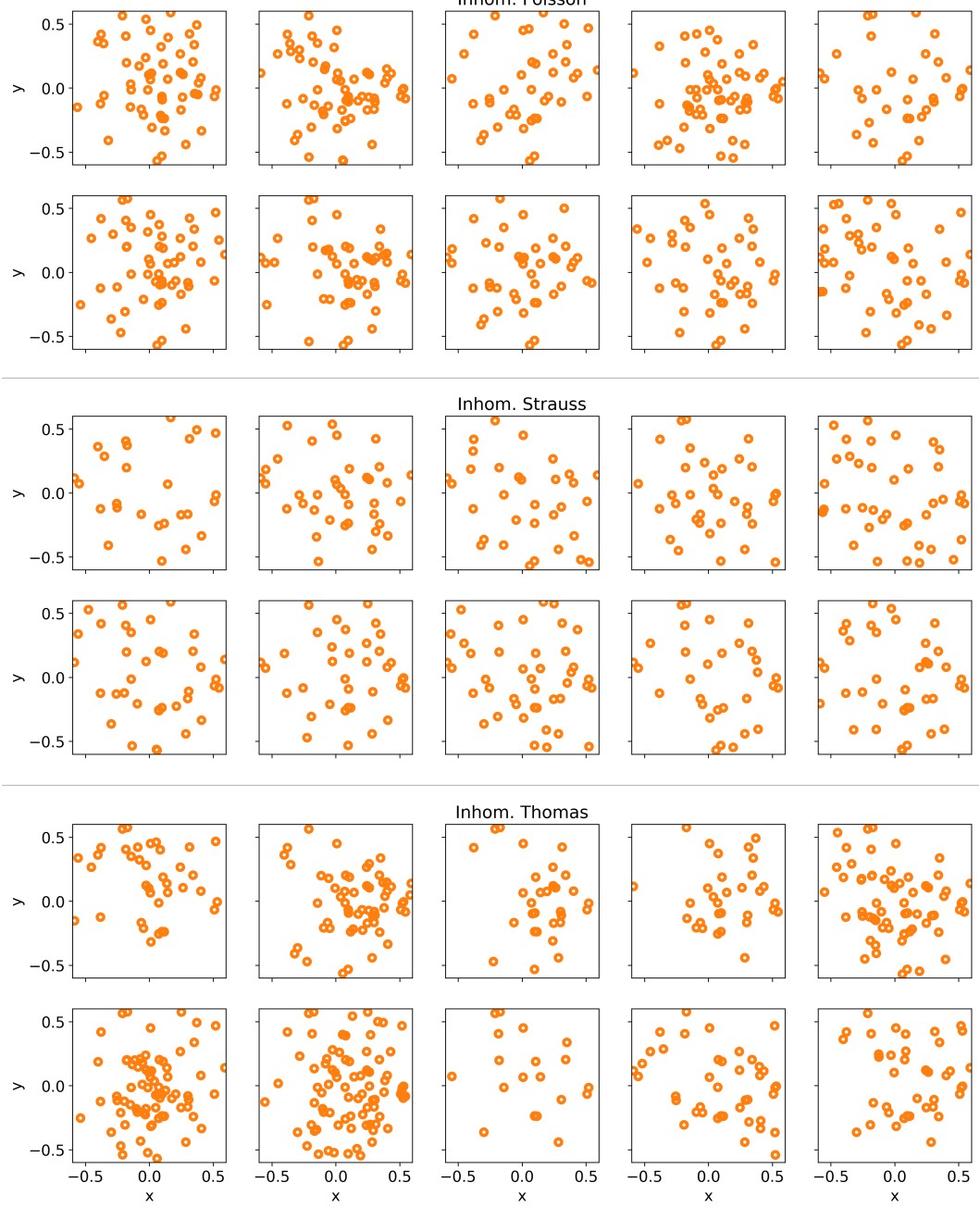

Figure 15: Examples of conditional simulation results from the trained conditional intensity model for different latent codes $z$ in the inhomogeneous cases over the evaluation domain $[-0.6, 0.6]^2$. The trained conditional intensity model is from the SI-VAE from Section 3.1, trained at high SNR with ground-truth knowledge. Ten *i.i.d.* samples are shown for each point process type (Poisson, Strauss, Thomas).

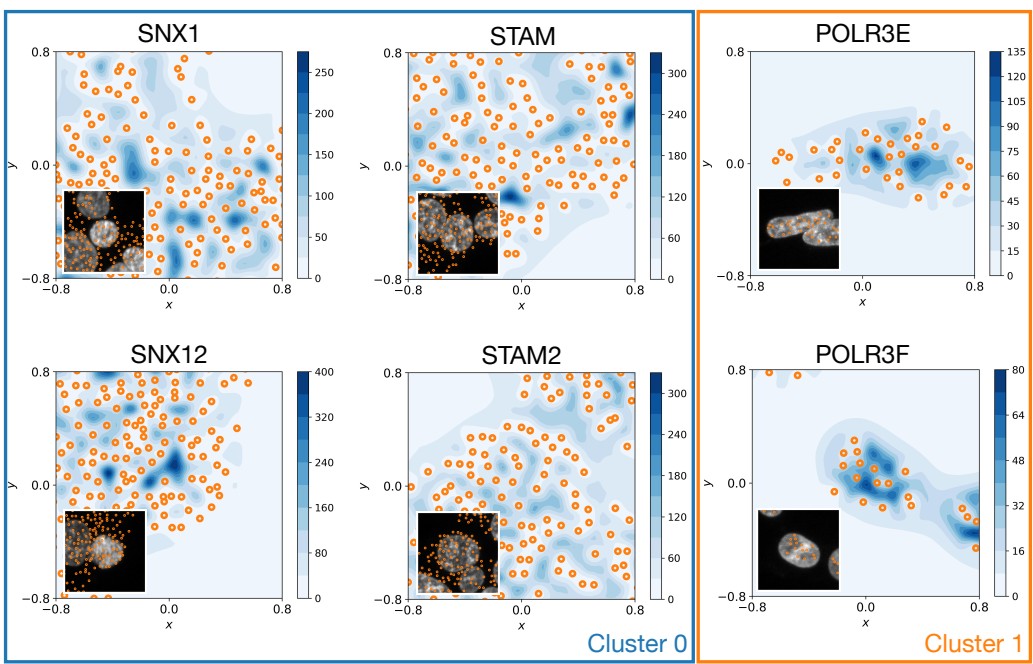

Figure 16: Conditional intensity predictions (contour plots) over the eroded domain $D = W \ominus R$ for representative samples from the test set of the *OpenCell* data with detected points in orange. The same samples as in Fig. 5 are shown with insets showing the corresponding nucleus channel for reference.

