# OpenReview forum: "Spatially Informed Autoencoders for Interpretable Visual Representation Learning"
_ICLR.cc/2026/Conference — ICLR 2026 Poster_

### Official Review · Reviewer_vPwX · 2025-10-29

**Soundness:** 3
**Presentation:** 2
**Contribution:** 3
**Rating:** 6
**Confidence:** 3

**Summary:**

The authors propose using spatial point processes as a self-supervision prior that explicitly models spatial distributions of objects to address the gap in previous un- and self-supervised methods that miss the spatial correlations.  Thus, the paper proposes spatially informed variational autoencoders (SI-VAE) to predict spatial organization patterns from images. The authors apply SI-VAE to a real world microscopy dataset, OpenCell, and correctly identify the protein localization classes.

**Strengths:**

- Improving VAEs by fusing them with spatial point processes, thus learning statistically interpretable representations of spatial distributions of objects in images can improve biological data analysis.

- Furthermore, the decomposition into interpretable potentials is also a non-trivial contribution that is based on well-established statistical frameworks.

- The paper introduces a principled way of interpreting the learnt representations within spatial statistics. The results on the biological dataset are promising.

**Weaknesses:**

- The real world applications were limited in the paper. Given the novelty of the proposed method, I would have wanted to see more results on biological data and also interpretations. While the results are promising, it is unclear to me whether SI-VAE generalizes to more complex localization patterns or proteins with overlapping spatial distributions.

- SI-VAE assumes that given z, the image and point pattern would be independent.
But since X is deterministically obtained from x, wouldn't this assumption be false? Could the authors clarify this?

**Questions:**

- Biological tissues can have direction dependent spatial structure (e.g.  muscle fibers, neuronal axons) which would lead to directionally correlated spatial patterns. Since SI-VAE assumes pairwise interaction potential to be symmetric, isotropic, from an implementation standpoint, how difficult would it be to extend SI-VAE to include anisotropic or direction aware interactions?

- How sensitive are the SI-VAE representations to errors in spot detection, and could the model be extended to be fully end-to-end (e.g. learning jointly the point locations and spatial interactions)?

---

> ### Author Response · Authors · 2025-11-21
> **Author response to review**
>
> We thank the Reviewer for taking the time to review our paper and for the spot-on summary of our contribution. We also thank the Reviewer for the constructive feedback regarding the weaknesses, which helped us strengthen the presentation in the revised paper.
>
> # Weakness 1
>
> Since we propose an entirely novel approach to spatial representation learning, we primarily aimed to evaluate the method in a principled, complete, and unbiased way. Therefore, the synthetic benchmarks are the main focus of the paper. This in particular tests the claims about the correlation structure of the representations and about conditional simulation; both would be difficult to evaluate on real data where ground truth is unknown. Nevertheless, we agree that showing practical applicability and usefulness is equally important, which is why we also included a real-world application case. We also agree that this was too short in the initial submission, mainly due to the page count limit. In the revised manuscript, we extended the real-world application along two directions: (1) We added two baselines -- VAE and the specialized state-of-the-art CytoSelf model -- and compare the learned representations and clustering performance with our approach. (2) We extended the discussion of the interpretation of the learned potentials to illustrate their scientific utility.
>
> # Weakness 2
>
> We thank the Reviewer for pointing out this point of confusion. The assumption that X and x are conditionally independent given z is trivially fulfilled in this particular setting. If X is a deterministic function of the image x (or vice versa), then z captures all relevant information about both of them. Hence, they are conditionally (on z) independent, as their cross-correlation function reduces to a Dirac delta. This is a very common assumption in multimodal VAE. Nevertheless, we made sure to include a better explanation of this in the revised manuscript.
>
> # Question 1
>
> We thank the Reviewer for this interesting question. Indeed, the current implementation of SI-VAE assumes isotropic pairwise interaction potentials. The model could be extended to anisotropic interactions by modifying the pairwise potential function to depend on the (signed) difference between points, i.e. $\psi(u,v)=\psi(u-v)$. As we show in Appendix B, learning such vector potentials is more challenging, but there is no fundamental limitation to do so within the SI-VAE framework. The main difficulty would likely be the availability of sufficiently diverse directional training data. This was clearly the problem in the experiments in Appendix B, since our current training data contain only isotropic interactions. In the revised manuscript, we added a discussion of this point and potential future work in the Conclusions section.
>
> # Question 2
>
> Noise robustness is indeed central to ensuring stable latent representations. We therefore intended to cover this point in Table 1 by comparing the downstream effects of different spot-detection methods (which generate different levels and types of errors and noise in the detections, see Appendix E.3). While, as expected, the performance of all methods deteriorates with increasing spot-detection noise, SI-VAE consistently outperforms the baselines. In the revised manuscript we now additionally report a sensitivity score (classification performance deterioration vs. spot-detection error) and discuss the practical implications. This now shows that the representations learned by SI-VAE dampen any error in the training data (spot locations are only required during training, not for inference) and do not amplify it.
>
> End-to-end learning of spot locations and spatial interactions is an interesting idea. However, it would require a more complex model architecture that combines spot detection with SI-VAE. It is not immediately clear then what the supervision target should be, since deep-learning spot-detection methods like Spotiflow require ground-truth annotations. If those are available, they could also be directly used to train an SI-VAE for which point locations are only required during training. Moreover, our results show that off-the-shelf pre-trained models such as Spotiflow provide sufficiently accurate spot detection for SI-VAE to learn accurate representations. Therefore, we believe that the current two-model approach is reasonable.

---

### Official Review · Reviewer_B6yH · 2025-10-31

**Soundness:** 2
**Presentation:** 3
**Contribution:** 2
**Rating:** 4
**Confidence:** 4

**Summary:**

This paper proposes a spatially informed variational autoencoder - SI-VAE, which consists of a VAE augmented with a spatial point process.  The latent representation z from the VAE is used as input to a neural network that then predicts the Gibbs potentials that define the point process.  Experiments are performed with synthetic data, showing a comparison to standard VAE, generalization to unseen processes, and zero-shot conditional simulation.  Lastly, an application to protein localization patterns is given, where it is demonstrated that the learned potentials agree with domain knowledge (eg proteins in vesicles being homogeneously distributed versus nucleus proteins being inhomogeneously distributed within the nuclei).

**Strengths:**

Interesting proposed model, sufficient technical contribution/novelty.  Validation on synthetic data.

**Weaknesses:**

Although the proposed model is interesting, majority of the validation is on synthetic data in some sense tuned to the specifics of the model.  Demonstration of applicability to a real problem is somewhat limited - consisting of only one specific test application where the final evaluation is a check that the learned model potentials agree at a high level with what is expected from domain knowledge.  Within this particular task, there is also no notion of a baseline for comparison.  This is an area where the impact could be much improved, by showing broader applicability to other domains, or giving a more qualitative analysis against baseline methods, or showing some unexpected/novel finding instead of confirming existing domain knowledge.

**Questions:**

see weaknesses above

---

> ### Author Response · Authors · 2025-11-21
> **Author response to review**
>
> We thank the Reviewer for appreciating the novelty of our proposed model and its validation on synthetic data. We also appreciate the constructive feedback on how to improve the real-world application example.
>
> Given the novelty of the SI-VAE framework, we primarily aimed to evaluate it in a principled, complete, and unbiased way. Therefore, the synthetic benchmarks are the main focus of the paper. This allows us to test the claims about capturing the correlation structure within the representations and of conditional simulation; both are difficult to quantitatively evaluate on real data where the ground truth is unknown. Our synthetic benchmarks are not tuned to the specifics of the model, but rather to validating the core claims of the paper in a controlled setting across all possible spatial processes (i.e., uncorrelated, attractive, repulsive, homogeneous, and inhomogeneous). Nevertheless, we agree that showing practical applicability and usefulness is equally important, which is why we also included a real-world application case. We also agree that this was too short in the initial submission, mainly due to the page count limit. In the revised manuscript, we extended the real-world application along two directions: (1) We added two baselines -- VAE and the specialized state-of-the-art CytoSelf model -- and compare the learned representations and clustering performance with our approach. (2) We extended the discussion of the learned potentials in order to illustrate how the interpretability of SI-VAE representations can be scientifically leveraged (which is not possible with either baseline).

---

### Official Review · Reviewer_zSUz · 2025-10-31

**Soundness:** 3
**Presentation:** 3
**Contribution:** 3
**Rating:** 8
**Confidence:** 2

**Summary:**

The paper developed a self-supervised deep-learning model  that use stochastic point processes to predict spatial organization patterns from images, coined as Spatially Informed Variational Autoencoders (SI-VAE). The self-supervision mechanism is modeled by the Papangelou conditional intensity. Extensive experiments were presented to illustrate the effectiveness of the SI-VAE model.

**Strengths:**

The paper provided a comprehensive illustration of the idea of Spatially Informed Variational Autoencoders (SI-VAE), which leverages the Papangelou conditional intensity as the self-supervision target for measuring the spatial information of images. Extensive experiments provided convincing results to showcase the effectiveness of the SI-VAE model in capturing spatial interactions and its generalization to unseen data in terms of zero-shot learning. The impact of the SI-VAE was also demonstrated on a challenging real-world application of protein localization in human cells.

**Weaknesses:**

Despite the strength as mentioned above, the paper only compares the proposed SI-VAE to the original VAE, while ignoring the existence of similar techniques, such as

1) Semenova, et al., PriorVAE: encoding spatial priors with variational autoencoders for small-area estimation, J R Soc Interface, 2022
2) Jazbec, et al., Scalable gaussian process variational autoencoders, AISTATS 2021.

Such an incompleteness of refereeing and comparisons weakens the overall quality of the paper.

**Questions:**

What are the connections between the present work with existing Gaussian process variational autoencoders? What is new in the SI-VAE? Does the current model promote easy implementations?

If Gaussian process variational autoencoders can also be used to capture spatial information in the images, can the SI-VAE model still outperform its Guassian process counterparts?

---

> ### Author Response · Authors · 2025-11-21
> **Author response to review**
>
> We thank the Reviewer for the appreciation of our work. The Reviewer correctly summarized the contribution of our work. We also thank the Reviewer for making us aware of additional relevant references. We cite them in the revised manuscript and discuss their relation with our work.
>
> Both papers mentioned by the Reviewer consider, as we do, the problem of learning correlations between spatial observations. Their approaches, however, differ fundamentally (i.e., in the underlying modeling premise) from ours:
>
> 1. The paper by Semenova et al. focuses on small-area estimation, which leverages neighborhood relations to improve estimation in areas with insufficient data. A VAE is used as a fast surrogate for Gaussian processes (GP) to enable efficient Bayesian inference. Therefore, the VAE is used to learn a low-dimensional representation of GP priors for Bayesian inference. In contrast, our work directly models the *interactions* between objects within images using spatial point processes. There, the point patterns themselves constitute the observations. The representations learned by the VAE thus predict the Gibbs potentials of the point process, rather than a Bayesian prior. This causes the latent representation of SI-VAE to be directly tied to the spatial structure of the data in the sense of a *hybrid model*. Moreover, while Semenova et al. argue that the latent representations learned by the VAE capture spatial correlations, we show that this is not necessarily the case (see Section 3.1 and 3.2). This motivates our choice to *directly* model correlations by enforcing $z$ to be a statistical predictor for the observed point distribution.
>
> 2. Jazbec et al. take the "opposite" approach. Instead of using a VAE to represent a GP prior, they use a GP *within* the VAE framework (GP-VAE) to model correlations between training data points in terms of the prior distribution over the latent $z$. Our notion of correlation refers to the interactions between objects within individual images, rather than between training images. Applying GP-VAE to our setting could improve the clustering of similar images over regular VAE, but it would not help model the spatial interactions of objects within individual images. In contrast, SI-VAE directly models these interactions by predicting the Gibbs potentials from the latent representations. This also makes the clusters mechanistically interpretable. We note, however, that combining both approaches could be an interesting direction for future work and added this to the Conclusions of the revised manuscript.
>
> Another difference between these previous works and SI-VAE is the model/computational complexity. SI-VAE are easy to implement and train, as they only require two small (just two layers) potential-prediction networks in addition to a regular VAE. The main additional cost during training is the evaluation of the Papangelou conditional intensity, which requires computing pairwise interactions between points within each image and numerical quadrature. However, this can be efficiently implemented using vectorized operations. During inference, SI-VAE have the same computational complexity as regular VAE; the point process component is only needed as self-supervision target during training. This makes SI-VAE a drop-in replacement for VAE in applications where spatial localization patterns are an important feature.

---

### Author Response · Authors · 2025-11-21
**Response to Reviews**

We thank the Reviewers for the careful reading of our manuscript and for the constructive comments. We have accounted for them in the revised manuscript, which we upload herewith. We provisionally upload a redlined revision of the manuscript to more easily spot the changes. This will be replaced with a clean version by Dec. 3 latest.

In addition, we provide point-by-point responses to all reviews below, explaining the revisions made. We invite the Reviewers to post answers or additional comments, if any, which we will gladly try to address by Dec. 3 to the extent possible.

---

> ### Author Response · Authors · 2025-12-03
> **Final revised manuscript uploaded**
>
> We regret there was no further discussion with the Reviewers. We do understand the circumstances, though. As announced, we replaced the redlined version of the revised manuscript with a clean version today, which we also proofread once more.
>
> We hope the AC appreciates our responses to the reviews and the revisions made to the manuscript. If a redlined markup simplifies things, we are happy to provide one upon request.
>
> Thanks!

---

### Meta-Review · Area_Chair_mwpY · 2026-01-06

**Summary:**

The reviews are overall positive (4, 6, 8). One of the larger weaknesses as pointed out by Reviewer B6yH (4) concerns lacking evaluations, which has however been addressed in the rebuttal through the addition of baselines. Overall, the manuscript merits acceptance.

Overall, the manuscript merits acceptance.

**Reviewer Concerns:**

See summary & reviewer scores. The authors clarified assumptions regarding conditional independence and demonstrated robustness to spot-detection errors.

**Reviewer Scores:**

Reviewer zSUz: 8 (unchanged)

Reviewer B6yH: 4 -> 5. The weak evaluation has been improved in the rebuttal with added baselines.

Reviewer vPwX: 6 -> 7. Some doubts were addressed in the rebuttal (e.g. independence assumption and real-world application)

---

### Decision · Program_Chairs · 2026-01-26

Accept (Poster)